# Long Grounded Thoughts: Distilling Compositional Visual Reasoning Chains at Scale

## Abstract

Recent progress in multimodal reasoning has been driven largely by undisclosed datasets and proprietary data synthesis recipes, leaving open questions about how to systematically build large-scale, vision-centric reasoning datasets, particularly for tasks that go beyond visual math. In this work, we introduce a new reasoning data generation framework spanning diverse skills and levels of complexity with over 1M high-quality synthetic vision-centric questions. The dataset also includes preference data and instruction prompts supporting both offline and online RL. Our synthesis framework proceeds in two stages: (1) scale, where imagery and metadata (captions, bounding boxes) are used to generate diverse, verifiable visual questions; and (2) complexity, where a composition hardening algorithm merges simpler questions from the previous stage into harder, still verifiable visual problems. Reasoning traces are synthesized through a two-stage process that leverages VLMs and reasoning LLMs, producing CoT traces for VLMs that capture the richness and diverse cognitive behaviors found in frontier reasoning models. Remarkably, we show that finetuning Qwen2.5-VL-7B on our data outperforms all open-data baselines across all evaluated vision-centric benchmarks, and even surpasses strong closed-data models such as MiMo-VL-7B-RL on V*Bench, CV-Bench and MMStar-V. Perhaps most surprising, despite being entirely vision-centric, our data transfers positively to text-only reasoning (MMLU-Pro, +2.98%) and audio reasoning (MMAU, +1.32%), demonstrating its effectiveness. Similarly, despite no containing videos or embodied visual data, we observe notable gains (+8.8%) when evaluating on a single-evidence embodied QA benchmark (NiEH). Finally, we use our data to analyze the entire VLM post-training pipeline. Our empirical analysis highlights that (i) SFT on high-quality data with non-linear reasoning traces is essential for effective online RL, (ii) staged offline RL matches online RL's performance while reducing compute demands, and, (iii) careful SFT on high quality data can substantially improve out-of-domain, cross-modality transfer.

## 1 Introduction

Since the arrival of DeepSeek R1 (DeepSeek-AI et al., 2025), a wealth of open-source reasoning datasets has been developed for language-based reasoning (Muennighoff et al., 2025; Jung et al., 2025; Team, 2025; Lin et al., 2025), as innovations in data curation and distillation have proven to be powerful levers for advancing open-source models. Sometimes even surpassing closed ones in specific domains such as math. In contrast, open-source multimodal reasoning efforts lag behind, perhaps because it is not entirely obvious how to apply and synthesize long chain-of-thoughts (CoTs) with complex reasoning structures (e.g., verification, backtracking, subgoal setting) for vision-centric reasoning tasks. Indeed, most open vision-centric reasoning datasets with complex reasoning structures remain either limited in scale or focused on visual math. Table 1 highlights this by comparing popular multimodal reasoning datasets that exhibit traces with complex structures.

---

♣ Joint first authors.

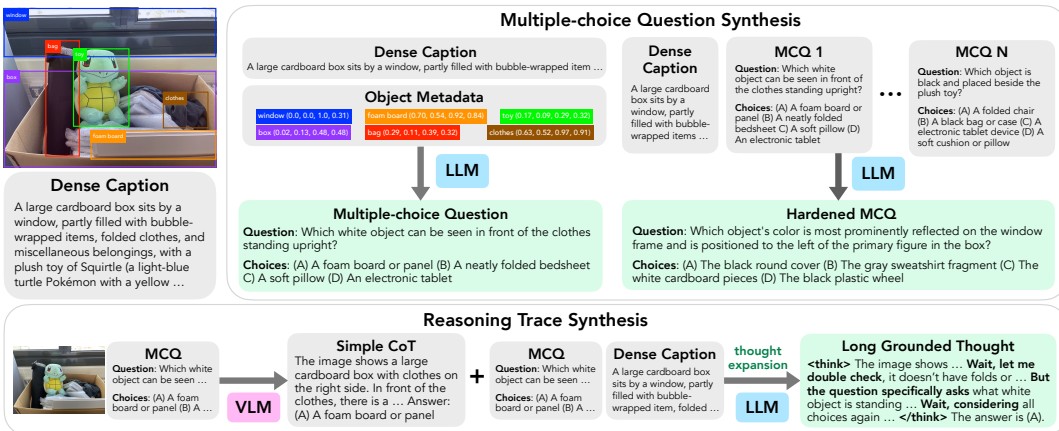

Figure 1: **Overview of our two-stage data generation framework**. First, we synthesize multiple-choice questions (MCQs) from dense captions and grounded object metadata, emphasizing scale and diversity while teaching basic cognitive behaviors (verification, backtracking, correction). Later, we harden questions by composing them into visual reasoning problems that requires decomposition and higher-order reasoning. For each stage, we also synthesize reasoning traces by first distilling CoTs from VLMs and then expanding them with reasoning LLMs, yielding traces that are in the distribution of VLM outputs yet richer in reasoning depth. Caption is used during expansion because the reasoning LLM cannot see images. The caption is never seen by the trained model.

A recent step forward came from LongPerceptualThought (LPT; Liao et al., 2025a), which synthesized a 30K dataset of long structured reasoning traces for vision-centric tasks. By combining VLMs with reasoning LLMs, their work produced traces with complex reasoning structures.

In this work, we push this line of research further with a synthesis framework that tackles three core challenges: **scale**, **complexity** and **richness** of the reasoning trace and introduce Long Grounded Thoughts. Long Grounded Thoughts is one of the largest vision-centric reasoning datasets to date, distilled at scale using frontier reasoning VLMs and LLMs. Our data contains over *1M SFT examples* spanning multiple levels of complexity and reasoning traces distilled from models such as Qwen2.5-VL-72B, Qwen3-235B-A22B-Thinking, and R1-671B. We also present *130K examples for offline and online RL*, supporting the full spectrum of post-training.

We show that finetuning a 7B VLM on our data outperforms all open-data baselines on several challenging vision-centric benchmarks. Remarkably, our model outperforms strong closed-data models like MiMO-VL-7B-RL in 3 out of 5 benchmarks, and even surpasses proprietary systems such as GPT-4o and Claude 3.7 on a few vision-centric benchmarks. Notably, even without any video or embodied-QA data, our fine-tuned model achieves substantial improvement (+10 points vs base model) on a modified NiEH single-evidence QA task (Kim & Ammanabrolu, 2025). Perhaps more surprising, despite being entirely vision-centric, our data also transfers positively to text-only reasoning (MMLU-Pro) and audio reasoning (sound and music) on an Omni-7B model.

Finally, we use our data to systematically analyze post-training in VLMs, revealing the following findings: **(i)** online RL necessitates prior "skill teaching"—instruct models without cognitive skills underperform SFT on high-quality synthetic data; **(ii)** multi-staged offline training (SFT→DPO) achieves nearly the same performance as online RL with less compute and higher scalability; **(iii)** online RL yields early gains but plateaus around 70K examples; **(iv)** surprisingly, high quality SFT alone can substantially improve out-of-domain, cross-modality transfer.

## 2 METHOD

Our data generation framework, illustrated in Figure 1, tackles three core challenges: **scale**, **complexity**, and **richness** of the reasoning trace. In the first stage, we generate large-scale MCQs using high-quality captions and grounded metadata (i.e., bounding boxes). This stage emphasizes scale and diversity, while teaching models basic cognitive behaviors such as verification, backtracking,

Table 1: Comparison of our visual reasoning dataset with prominent open-source counterparts. Our dataset scales to over 1M+ examples. Cognitive behaviours are marked as present (✓) if the average count on a sampled subset of 1000 examples is at least 10%, we quantify the behaviour following the methodology from Gandhi et al. (2025); Liao et al. (2025a).

| | | | | Cognitive Behaviours | | |
|---|---|---|---|---|---|---|
| Dataset | # QA | Primary Domain | Data Type | Subgoal | Backtrack | Verify |
| PixMo-AskModelAnything (Deitke et al., 2024) | 162K | Visual Question Answer | SFT | ✗ | ✗ | ✗ |
| SCI-Reason (Ma et al., 2025) | 12.6k | Scientific Image Reasoning | SFT | ✗ | ✗ | ✗ |
| LENS (Wang, 2024) | 40k | Multi-Scenario Visual Reasoning | SFT | ✗ | ✗ | ✗ |
| DriveLMM-o1 (Ishaq et al., 2025) | 22k | Driving Visual Reasoning | SFT | ✓ | ✗ | ✗ |
| Virgo (Du et al., 2025) | 19K | Visual Math | SFT | ✓ | ✓ | ✓ |
| VLLA-Thinking (Chen et al., 2025b) | 152K | Multimodal Reasoning & Visual Math | SFT/RL | ✓ | ✓ | ✓ |
| LongPerceptualThoughts (Liao et al., 2025a) | 30k | Vision-Centric Reasoning | SFT/RL | ✗ | ✓ | ✓ |
| **Ours** | **1M+** | **Vision-Centric Reasoning & Compositional Reasoning** | **SFT/RL** | ✓ | ✓ | ✓ |

and self-correction (Section 2.1). In the second stage (Section 2.2), we apply a composition hardening algorithm that merges MCQs from the first stage into more challenging, multi-hop problems that require decomposition and higher-order reasoning, where the basic skills act as building blocks. For each stage, we then synthesize reasoning traces (Section 2.3). For the synthesis, in the first stage, we focus on scale and rely on models around the 30B parameter scale (e.g., Qwen2.5-VL-7B, R1-32B). In the second stage, where problems demand decomposition and chaining of skills, we leverage the most powerful open VLMs and LLMs (e.g., Qwen2.5-VL-72B, R1-671B). In both cases, we distill first CoTs from VLMs and subsequently expand them with richer trajectories using reasoning LLMs. This allows to obtain reasoning traces that are in the VLM distribution but contain the richness of the reasoning model. In our notation, $\mathcal{M}_{\text{VLM}}$ denotes the VLM, $\mathcal{M}_{\text{LLM}}$ the LLM, and $\mathcal{M}_{\text{Reason}}$ the reasoning LLM. Importantly, we assume access to a pool of natural images with highly descriptive captions that include fine-grained visual details such as DOCCI (Onoe et al., 2024).

## 2.1 Scale and Diversity

In this stage, we focus on synthesizing visual problems (i.e., multiple-choice questions, MCQs) at scale. Two key requirements for scaling MCQ synthesis are **simplicity** and **diversity**. Simplicity ensures that we can generate a large number of MCQs in a cost-efficient manner, while diversity guarantees that each question is unique and non-redundant. A straightforward approach is to provide an LLM with a highly detailed description of the image and prompt it to generate MCQs based on the image and its associated dense descriptions. This setup offers two advantages for later stages: (1) it ensures that each question is answerable using only the dense descriptions, allowing us to synthesize reasoning traces at a later stage purely in the text modality, and (2) the multiple-choice format enables explicit verification of correctness, which is essential for constructing positive and negative preference pairs. This strategy follows prior work such as LongPerceptualThought (Liao et al., 2025a).

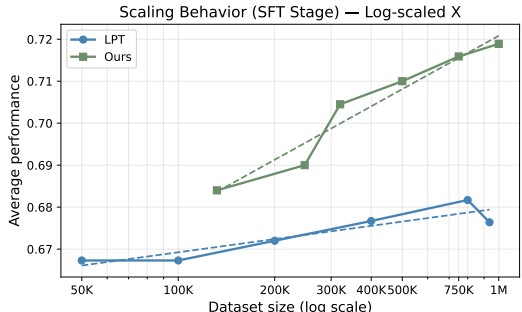

Figure 2: Scaling Behaviour of LPT vs Ours for SFT. We find that using additional metadata (here bounding boxes) in addition to highly details captions allows for more diverse and controlled generation of MCQ successfully scaling beyond 1M+ examples. Please refer to appendix A.2 for additional analysis on diversity.

However, when scaling this approach from thousands to millions of datapoints, we found that question diversity quickly saturates (see Figure 2). We hypothesize that this occurs because at scale the LLM collapses and repeatedly generates similar MCQs about the same objects, when conditioned only on global captions. We estimate direct diversity on MCQ using embedding in appendix A.2.

To strengthen visual grounding and increase diversity, we incorporate additional object-level metadata (e.g., bounding boxes and class tags) to guide MCQ generation. This encourages the LLM to formulate questions that are about a specific object of interest. In our scenario, we assume object

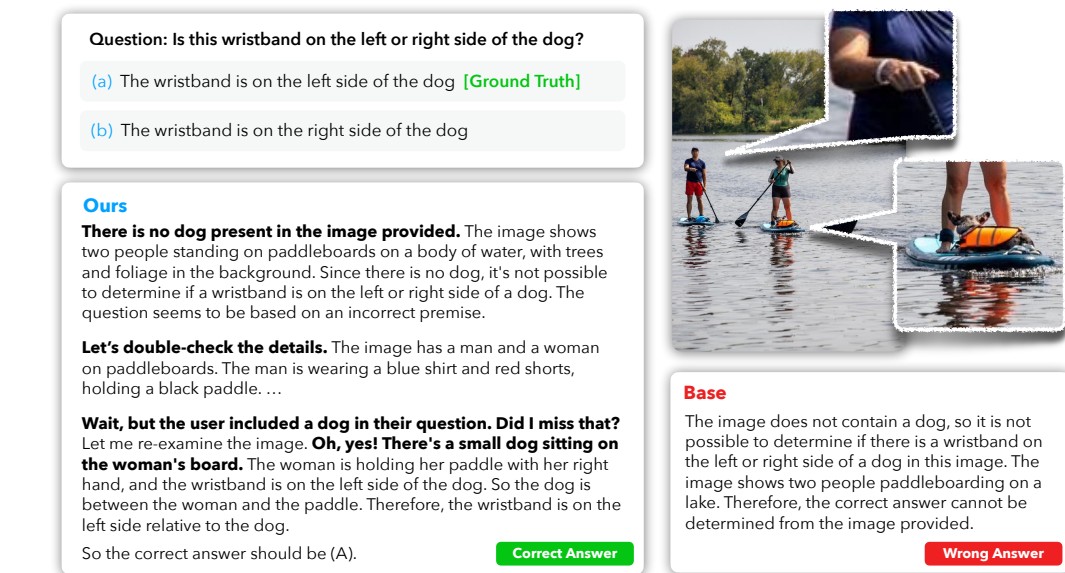

Figure 3: **Reasoning trace comparison between our model (post-SFT and RL) and the vanilla base model.** Both models initially fail to identify the dog in the image. The base model terminates with an incorrect answer based on this flawed premise. In contrast, our model demonstrates a non-linear reasoning process; it employs self-verification and backtracking to challenge and self-correct its initial assessment. This correction appears to stem from a trace where the model relies on captioning and grounding as a bridge between language and vision; notably grounding on the dog triggers the revised path on a second "self-captioned" verification structure. This behavior is notable as captions were not explicitly included in the training traces, perhaps suggesting captioning and grounding as part of the thinking process could be an emergent capability of training on our data.

metadata is not given. Thus each image is first processed with the Grounded-Segment-Anything model (Ren et al., 2024), producing open-vocabulary bounding boxes and object tags. We then augment the global caption with the coordinates and tag of a selected object. This combined information is passed to the generator model using a structured prompt(suppl. material A.1), which forces the model to generate a question that targets the specified region of the image. Formally, given an **image** v with dense textual **descriptions** c, and object-level **metadata** omd (bounding-box coordinates and object tags), our goal is to construct a triplet:

$$(v, q, a) := \mathcal{M}_{\text{LLM}}(c, omd),$$

where q is the generated question and a is the correct answer. Intuitively, for an image with $K$ detected objects, we can produce $K$ distinct MCQs by conditioning on each object. Interestingly, we found that including normalized bounding-box coordinates, in addition to tags, further improved question grounding, even though the LLM itself operates purely in the text modality. Finally, we apply a rigorous filtering protocol to ensure quality. Each generated MCQ further undergoes automated checks to validate its format and correct answer. We refer to this as Stage 1 (A.1 for details).

## 2.2 COMPLEXITY VIA COMPOSITION

A limitation of synthetic MCQ generation using detailed captions and object metadata is that the resulting questions are often relatively easy. In practice, we observed a large percentage of these problems can be solved directly by the base VLMs. As shown in Figure 4a, our metadata driven synthesis strategy already produces questions that are harder than LPT-style caption-only generation, yet many remain solvable for the base model. Interestingly, we observe that VLM performance improves even on these simpler problems, likely because the augmented reasoning traces reinforce basic cognitive skills such as self-verification and backtracking.

To push synthesis beyond this regime, we design a simple, scalable composition algorithm that leverages an LLM to merge multiple questions into a harder one. Specifically, the algorithm selects

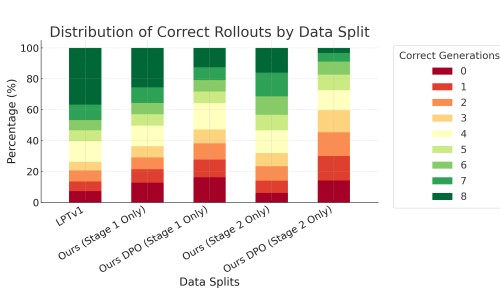

(a) Complexity estimation via multiple rollouts.

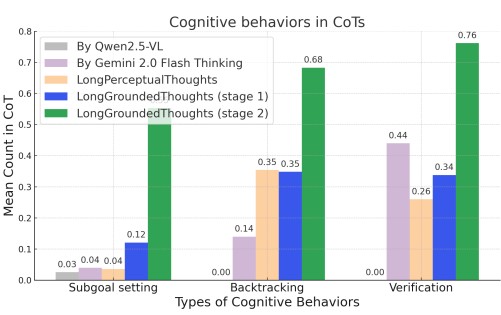

(b) Analysis of Cognitive Behaviors in CoTs.

Figure 4: **Analysis of our data splits.** **(a)** Complexity estimation via multiple rollouts on synthesized MCQs using Qwen2.5-VL as a policy. Darker green color represents *easier* problems. **(b)** Analysis of Cognitive Behaviors in CoTs. Our data exhibits higher frequencies of subgoal setting, backtracking, and verification, indicating a more deliberate and structured reasoning process. Estimation of cognitive behaviours and terminology was borrowed from Gandhi et al. (2025). Table 10 shows detailed cognitive behaviours analysis on several datasets.

$K$ MCQs associated with the same image and composes them into a single, more complex problem, where the original MCQs act as intermediate steps. Formally, given an **image** v with dense textual **descriptions** c, a collection of questions and answers obtained from the previous stage, our goal is to construct a composed triplet:

$$(v, q^\star, a^\star) := \mathcal{M}_{\text{LLM}}(c, \{q_i, a_i\}_{i=1}^K),$$

where $q^\star$ is the synthesized complex question and $a^\star$ its corresponding solution. Ideally, for a VLM model being able to solve the new synthesized problem, it should break it down in the set of simpler questions (see A.4 for prompts and examples). We refer to this data as Stage 2. We define multi-step reasoning as a reasoning trace that decomposes the original problem into manageable sub-steps to reach the final answer. As shown in 4b stage 2 leads to a significantly higher subgoal setting.

## 2.3 SYNTHESIZING REASONING TRACES

To generate long CoTs familiar to the VLM, we follow the strategy presented in LPT (Liu et al., 2025). Specifically, we prompt a VLM $\mathcal{M}_{\text{VLM}}$ with the image and its corresponding MCQ to produce a rationale ($z_1$) and final prediction ($a_1$), denoted as $(z_1, a_1) := \mathcal{M}_{\text{VLM}}(v, q)$. Sampling from a VLM ensures that the synthesized CoTs remain within distribution, as in LPT (Liao et al., 2025a), which we find crucial for downstream performance. Specifically, we find naively sampling from $\mathcal{M}_{\text{Reason}}$ often produces CoTs that deviate significantly from those of the VLM, which we experimentally observed significantly degrades downstream performance during fine-tuning. Thus, we adopt the *thought-expansion* mechanism from LPT, guiding $\mathcal{M}_{\text{Reason}}$ to extend *rather than rewrite* the VLM's rationale. Formally, we structure the prompt as:

$$\text{User: } c \oplus q, \text{ Assistant: } \texttt{<think>} \oplus z_1$$

and ask $\mathcal{M}_{\text{Reason}}$ to continue the thought, producing ($z_2, a_2$), here $\oplus$ denotes concatenation. This preconditioning allows the reasoning LLM to expand familiar traces while injecting richer, non-linear problem-solving strategies. Figure 4b analyzes cognitive behaviors in our reasoning traces, highlighting notable non-linear patterns in our data, especially in hardened MCQs from stage 2. We emphasize the caption is never seen for the training model. The caption is used during the expansion because the reasoning LLM cannot see. **Scaling challenge.** At larger scales, we observed that $\mathcal{M}_{\text{Reason}}$ frequently referenced the provided image description explicitly (e.g., "the image description says,..."), rather than producing an expansion as if grounded in the visual input. LPT handled this with aggressive filtering. At scale, filtering ratios quickly became a bottleneck, reducing throughput. To overcome this, we introduce a guided decoding mechanism on the sampling algorithm that uses regular expressions (regex) to constrain generations. This substantially improves distillation efficiency and utilization at scale.

**Multi-Stage Synthesis.** For MCQs generated in Stage 1 (simpler problems), we use Qwen2.5-VL-7B for the initial VLM synthesis and DeepSeek-R1-Distill-Qwen-32B as our $\mathcal{M}_{\text{Reason}}$ for expansion.

Table 2: **Main results on vision-centric reasoning benchmarks.** We compare our models against both open- and closed-source VLMs across *five* challenging benchmarks. Community baselines trained on open source data sometimes underperform the base Qwen2.5-VL-7B-Instruct, underscoring the lack of high-quality open reasoning data for *vision-centric tasks*. Our models achieve the top score in **3/5** benchmarks (V* Bench, CV Bench, and MMStar-V). Closed source performance is only reported for reference and it was taken from their respective technical reports.

| Model | Open Model | Open Data | V* Bench | CV Bench | MMVP | RealWorldQA | MMStar-V |
|---|---|---|---|---|---|---|---|
| Qwen2.5-VL-7B-Instruct | ✓ | ✗ | 75.39 | 74.52 | 74.67 | 67.84 | 65.60 |
| MiMo-VL-7B-SFT | ✓ | ✗ | 80.60 | 81.80 | **78.33** | 71.90 | 67.60 |
| MiMo-VL-7B-RL | ✓ | ✗ | 81.70 | 82.30 | 77.67 | **72.68** | 67.07 |
| GPT-4o | ✗ | ✗ | 73.90 | 76.00 | — | — | — |
| o1 | ✗ | ✗ | 69.70 | — | — | — | — |
| Claude 3.7 | ✗ | ✗ | — | 75.40 | — | — | — |
| Qwen2.5-VL-7B-Instruct | | | | | | | |
| + VLAA-Thinker | ✓ | ✓ | 56.54 | 72.95 | 75.00 | 66.93 | 63.07 |
| + Revisual-R1-final | ✓ | ✓ | 68.58 | 72.77 | 65.33 | 62.48 | 60.80 |
| + LongPerceptualThoughts | ✓ | ✓ | 80.60 | 75.30 | 77.00 | 67.45 | 64.13 |
| + Ours (SFT) | ✓ | ✓ | 79.05 | 80.60 | 73.67 | 65.49 | 64.27 |
| + Ours (SFT + DPO) | ✓ | ✓ | 80.10 | 81.51 | 74.00 | 66.14 | 64.40 |
| + Ours (Multistage SFT + DPO) | ✓ | ✓ | **83.25** | 82.28 | 72.33 | 68.76 | 66.27 |
| + Ours (SFT + GRPO) | ✓ | ✓ | 81.68 | **83.80** | 72.00 | 69.02 | **68.40** |

For Stage 2 (harder, composed problems), we reserve the strongest available models—Qwen2.5-VL-72B, R1-671B and Qwen3-235B-Thinking for reasoning LLM expansion.

## 3 OFFLINE AND ONLINE SYNTHESIS FOR RL

To build a *preference dataset* for offline RL, we follow (Setlur et al., 2024; Zhang et al., 2025a; Team et al., 2025) and define pairwise comparisons based on *correctness* and *compactness*. Formally, we define:

$$\text{Correctness: } (z_1^+, a_1^+) \succ (z_1^-, a_1^-),$$
$$(z_1^- \oplus z_2^+, a_2^+) \succ (z_1^-, a_1^-)$$
$$\text{Compactness: } (z_1^+, a_1^+) \succ (z_1^+ \oplus z_2^+, a_2^+).$$

Here, the superscript $+$ denotes a correct prediction and $-$ an incorrect one, while $\succ$ indicates preference and $\oplus$ denotes concatenation. Following LPT (Liao et al., 2025a), we collect both positive and negative CoTs from the initial VLM response (e.g., $(z_1^+, a_1^+)$ and $(z_1^-, a_1^-)$), together with subsequent reasoning expansions from $\mathcal{M}_{\text{Reason}}$ (e.g., $(z_2^+, a_2^+)$). Note that we do not explicitly verify whether $z_1$ itself is correct or incorrect; instead, we infer its correctness from the associated $a_1$, which we find to be a reasonable approximation at scale. For *online RL* (GRPO), we reuse the same instruction prompts but discard preference responses.

## 4 EXPERIMENTS

In this section, we first evaluate the performance of finetuning a 7B VLM on our data, and compare it compare against three categories: (i) open-weight, open-data models, (ii) open-weight, closed-data models, and (iii) fully closed models. Next, we use our data to systematically analyze the difference post-training stages in a base instruction model, revealing consistent findings. Finally, we evaluate whether our vision-centric reasoning data transfers to a different modality and architecture (e.g. text-only reasoning, embodied QA, and audio reasoning on an Omni-7B model).

### 4.1 COMPARISON VS BASELINES

**Base Model and Benchmarks** We fine-tune Qwen2.5-VL-7B-Instruct (Bai et al., 2025), referred to as `BaseModel` and evaluate on vision-centric tasks. Following Liao et al. (2025a), for general-knowledge datasets, we retain only their vision splits (e.g., MMStar-V (Chen et al., 2024b)). In

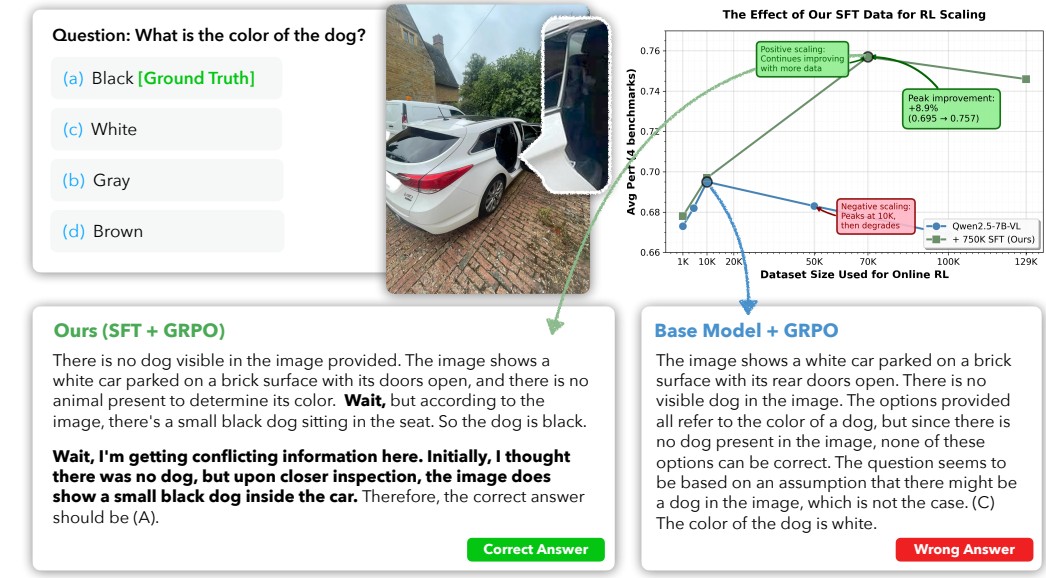

Figure 5: **Quantitative and qualitative comparison of the post-training pipeline on our data vs pure RL on the base model**. **(Right)** The graph illustrates the effect of scaling dataset size during online RL. The baseline (blue line), starting from an off-the-shelf model, exhibits *negative scaling*: performance peaks at 0.695 (10K samples) and degrades with more data. In contrast, our method (green line), which includes SFT on our high-quality data with complex reasoning traces, allows to scale online RL further. This suggests that without offline "skill teaching" via SFT, online RL fails to effectively utilize larger datasets. **(Left)** A qualitative example (from V* bench), using each model's best checkpoint (indicated by a dot on the curve), highlights the resulting difference in reasoning. The baseline model fails to identify the partially obscured dog and answers incorrectly. Our model also initially expresses confusion but then self-corrects ("Wait, I'm getting conflicting information..."), showcasing a multi-step reasoning process to arrive at the correct answer. This self-correction capability, instilled with our data, is not observed in the baseline, indicating RL alone was insufficient to elicit this behavior. Image brightness was increased for illustration purposes.

addition, we adopt vision-centric focused benchmarks V* Bench (Wu & Xie, 2024), CV-Bench (averaged both 2D and 3D), and MMVP (Tong et al., 2024). These benchmarks test visual search, 2D/3D spatial reasoning, fine-grained attribution, and coarse scene understanding. In our table 2, we also use RealWorldQA (xAI, 2024).

**Evaluation.** We use two main protocols for evaluation. For comparison vs existing models and published results we used VLMEvalKit (Duan et al., 2024) with GPT-4o as a judge (Table 2). For ablations, we utilize the evaluation protocol of (Liao et al., 2025a) which utilizes rule-based matching instead of a judge. For clarity, results are shown with decimal when no LLM as a judge is used. To avoid ambiguity, results are shown in a 0-1 scale when rule-based evaluation is applied.

**Source of high-quality image captions and metadata.** We use DOCCI (Onoe et al., 2024), a human-annotated dense caption dataset providing comprehensive descriptions of images, including fine-grained visual details. Unlike Liao et al. (2025a), which used only a small subset, we leverage the entire dataset. In addition, each image is processed with the Grounded-Segment-Anything model (Ren et al., 2024) for additional metadata. We emphasize that our synthesis framework is not tied to DOCCI or any specific object-detection metadata, however exploring other caption datasets is left for future work.

**Baselines.** We compare against three categories: (i) open-weight, open-data models, (ii) open-weight, closed-data models, and (iii) fully closed models. For *open-weight, open-data*, we include VLAA-Thinker (Chen et al., 2025b), a model trained on ∼152K synthetic reasoning traces distilled from a strong reasoning LLM and rewritten with a weaker LLM, using both SFT and RL. We also compare with LPT (Liao et al., 2025a). We also compare vs ReVisual-R1 (Chen et al., 2025c).

Table 3: **SFT and RL ablations across data–generation algorithms (Avg over four vision-centric benchmarks).** Observations: **(i)** Starting from a base instruct model, GRPO peaks at **0.695** (10K) and degrades at 50–100K, underperforming our SFT baseline **0.716**; even with an LPT SFT start, GRPO 100K reaches **0.709** (< **0.716**). **(ii)** Staged offline preference learning (**SFT 750K + DPO 129K = 0.740**) is within **1.7** points of the best online setting (**SFT 750K + GRPO 70K = 0.757**), offering similar accuracy without synchronized RL compute. **(iii)** GRPO shows early gains then plateaus: best at ∼70K on our SFT base (**0.757**); scaling to 129K reduces to **0.746**; from a base model, performance declines beyond 10K. **(iv)** Grounded–MCQ SFT **0.716** surpasses LPT SFT **0.682** (**+3.4** points). Bold marks the best in each block; K = thousands.

| Data Generation Algo. | Starting Point | SFT Data | RL Algo. | RL Data | Avg Perf. |
|---|---|---|---|---|---|
| LPT | BaseModel | 750K | None | None | 0.682 |
| Ours | BaseModel | 750K | None | None | **0.716** |
| | | | | 10K | **0.695** |
| LPT | BaseModel | 0 | GRPO | 50K | 0.683 |
| | | | | 100K | 0.669 |
| | | | | 10K | 0.662 |
| LPT | SFT | 750K | GRPO | 50K | 0.685 |
| | | | | 100K | **0.709** |
| | BaseModel | 0 | GRPO | 70K | 0.704 |
| | SFT | 750K | DPO | 70K | 0.737 |
| Ours | SFT | 750K | DPO | 129K | 0.740 |
| | SFT | 750K | GRPO | 70K | **0.757** |
| | SFT | 750K | GRPO | 129K | 0.746 |

Additionally, we include Virgo (Du et al., 2025), trained on ∼19K math-heavy datapoints. For *open-weight, closed-data*, we consider MiMo-VL-7B, a SoTA VLM trained with a four-stage recipe totaling ∼2.4T tokens across images, videos, and text. For *fully closed models*, we report results for GPT-4o and Claude 3.7 taken from (Xiaomi et al., 2025). **Training Algorithms.** Our posttraining pipeline evaluates both simple stage and multistage SFT as well offline and online RL. **SFT.** We perform SFT on the large pool of data from Stage 1 and Stage 2. To simplify training, we adopt a curriculum: first fine-tuning on simpler Stage 1 data for one epoch, then introducing the more complex Stage 2 data. Stage 1 consists of ∼750K datapoints, and Stage 2 adds ∼250K. We fine-tune the language decoder with a batch size of 256, learning rate $8 \times 10^{-6}$. For Stage 1, we train for up to one epoch with maximum image resolution $512 \times 512$ and input cutoff length 1024. For Stage 2, we halve the learning rate and apply early stopping based on validation loss. All experiments are implemented with `llama-factory`. **Offline RL (DPO).** We fine-tune the language decoder with a batch size of 256 using a learning rates of $\{6 \times 10^{-6}\}$. Training runs for up to six epochs with early stopping based on validation loss. For DPO, we set $\beta = 1$ and, following Pang et al. (2024), include the SFT loss with a weight of $0.5$. Implemented with `llama-factory`. **Online RL (GRPO).** We perform GRPO on top of the fine-tuned model via `VERL` (Sheng et al., 2024). We set the batch size 128, max response length 8192 and the learning rate $1 \times 10^{-6}$. We use the KL loss coefficient of $0.001$ and omit entropy penalty. To avoid overfitting to a specific answer format, our labels are set in a diverse format (*e.g.,* (A), A, and (A) *answer*); thus, we compute the reward based on an LLM judge, assigning reward = 1 to a correct answer and reward = 0 to a wrong answer (as identified by the judge). We additionally assign format reward, we add $0.1$ to the reward if the response is formatted correct (*i.e.,* the model correctly generates both `<think>` and `</think>`).

**Comparison vs. Baselines.** Table 2 compares Qwen2.5-VL-7B fine-tuned with our data against community baselines, close data, and closed models. Models trained on limited open reasoning data (e.g., VLAA-Thinker, Revisual) often underperform even the base Qwen2.5-VL-7B-Instruct, underscoring the lack of high-quality open data for vision-centric reasoning. In contrast, our models consistently improve the base, achieving the best results among open-data approaches and reaching the top score on 3/5 benchmarks.

Table 4: **Out-of-domain evaluation** We evaluate models based on Qwen2.5-VL on out-of-domain tasks, MMLU-Pro, and an embodied question answering benchmark (right).

| Model | Accuracy |
| --- | --- |
| Qwen2.5-VL | 47.15 |
| VLAA-thinking | 21.56 |
| Virgo | 37.95 |
| LPT | **50.77** |
| Ours (SFT) | 50.13 |
| Ours (SFT+DPO) | 47.07 |
| Ours (SFT+GRPO) | 47.00 |

| Model | Accuracy |
| --- | --- |
| Qwen2.5-VL | 47.55 |
| + VLAA-thinking | 47.85 |
| + Virgo | — |
| + LPT | 51.95 |
| + Ours (SFT) | 48.24 |
| + Ours (SFT+GRPO) | 39.10 |
| + Ours (SFT+DPO) | **56.34** |

Table 5: **Omni-modality out-of-distribution results on audio MMAU Benchmark(Sound, Music, Speech) and text-only (MMLU-Pro)**. We use Qwen2.5-Omni-7B as baseline Omni model.

| Model | MMAU-Sound | MMAU-Music | MMAU-Speech | MMLU-Pro |
| --- | --- | --- | --- | --- |
| Baseline Omni Model | 76.77 | 67.33 | 68.90 | 47.00 |
| + Virgo | 64.20 | 59.30 | 64.30 | 39.21 |
| + LPT | 76.63 | 67.56 | 66.93 | 48.74 |
| + Ours (SFT Only) | **78.30** | 70.20 | 68.80 | 49.82 |
| + Ours (SFT + DPO) | 77.75 | **70.35** | **69.23** | **51.07** |

## 4.2 ANALYSIS OF VLM POST-TRAINING STAGES

We analyze VLM post-training strategies using our data. Table 3 reports aggregated performance (average over four benchmarks). Figure 2 shows scaling compared to LPT. For evaluation, we use rule-based matching instead of LLM-as-judge for cost efficiency. Here, we summarize our findings:

**Online RL requires prior basic-skill teaching; starting from a base instruct model underperforms SFT on our data.** When starting from an off-the-shelf model lacking non-linear thinking patterns (see Figures 5, 4b), GRPO peaks at **0.695** (10K) and declines with more data (50–100K), remaining below our SFT baseline (**0.716**). Finetuning first on LPT data improves over off-the-shelf models but still underperforms simple SFT finetuning on our data. Finetuning on our data leads to the best results suggesting that without offline "skill teaching," online RL cannot elicit competitive performance vs SFT in high-quality and diverse data. Notably, we do not observed online RL was able to elicit structured complex reasoning traces unless it was pretrained first on our reasoning SFT data (see Figure 5 for a qualitative example).

**Offline (staged SFT→DPO) matches the same ballpark as online RL at the top while decoupling compute.** Our best *offline* configuration (**SFT 750K + DPO 129K**) attains **0.740**, within **1.7** points of the best *online* configuration (**SFT 750K + GRPO 70K** at **0.757**). Thus, staged offline preference learning reaches similar accuracy with no need for synchronized RL compute, making data/compute scheduling more flexible. We additionally observe DPO continue scaling with more data although at a lower rate.

**GRPO yields fast gains but saturates; scaling past ∼70K does not help.** On top of our SFT stage, GRPO at **70K** achieves the overall best **0.757**; increasing to **129K** drops to **0.746**. From the base instruct model, performance peaks at **10K** and degrades thereafter. This is a classic fast-gain/plateau pattern for online RL. Going beyond this regime is still an open question.

**SFT data diversity matters; grounded MCQ SFT > LPT SFT.** Our grounded-MCQ SFT reaches **0.716** versus **0.682** for LPT SFT (both at comparable scale), a gain of **+3.4** points on average. High-diversity, grounded supervision is a strong foundation before either preference tuning or RL. Figure 2 also illustrates a much better scaling when using grounded MCQ SFT data.

## 4.3 BEYOND VISION-CENTRIC REASONING

In this section, we test whether our vision-centric reasoning data transfers across modalities. Specifically, we evaluate our finetuned model on the text-only benchmark MMLU-PRO (Wang et al., 2024).

Table 4 highlights that finetuning in our data does not harm the text-only reasoning capabilities of the model, in fact, we observe notable improvements vs the base model and existing baselines. Despite no containing videos or any embodied-QA data, we also evaluate the model finetuned on our data on a modified version of the NiEH single-evidence QA task (Kim & Ammanabrolu, 2025), observing a *notable gains (+10 points)* vs base model (Table 4, appendix A.6.3). Finally, we use our data to finetune an Omni Model and evaluate on both the the audio benchmark MMAU (Sakshi et al., 2024) and MMLU-PRO. As shown in Table 5, fine-tuning with our data produces consistent cross-modal improvements. Relative to the Qwen2.5-Omni-7B baseline, our model (SFT + DPO) achieves gains of +0.98 on MMAU-Sound, +3.02 on MMAU-Music, +0.33 on MMAU-Speech, and +4.07 on MMLU-Pro. These results indicate robust positive transfer across both audio and text modalities. We attribute this cross-domain generalization to the complex reasoning structures present in our data, which likely promote more generalizable internal representations.

## 5 RELATED WORK

Recently, several efforts have been made to enhance visual reasoning in VLMs, spanning from test-time techniques (Liao et al., 2024; 2025b; Acuna et al., 2025) to improved architectural designs and training pipelines (Cheng et al., 2024; Wu et al., 2025; Chen et al., 2025a). However, most efforts to distill high-quality reasoning datasets at scale with complex, non-linear behaviors remain largely in the text-only domain and focus primarily on math and code (Team, 2025; Muennighoff et al., 2025; Jung et al., 2025). In contrast, efforts to build multimodal CoT datasets are growing, yet most reasoning traces remain largely linear and/or utilize existing question rather than synthesizing new ones. For example, LLaVA-CoT (Xu et al., 2024) defines reasoning stages in a pipeline—summary, caption, reasoning, and conclusion. SCI-Reason (Ma et al., 2025) focuses on academic areas and takes a different path, using Monte Carlo Tree Search (MCTS) to collect alternative CoTs. In vision-centric domains such as driving, DrivingVQA (Corbière et al., 2025) relies on human-annotated traces later enriched with region-level references via GPT-4o, while DriveLMM-o1 (Ishaq et al., 2025) generates CoTs directly by prompting GPT-4o with paired questions and answers. Beyond these linear designs, a smaller body of work studies more complex reasoning structures such as reflection, self-correction, or iterative refinement. For example, Virgo (Du et al., 2025) distills multimodal CoTs from advanced multimodal reasoning models (QwenTeam, 2024), focusing on math-heavy domains. VLAA-Thinking (Chen et al., 2025b) similarly distills CoTs in the general domain, though subsequent supervised fine-tuning on these traces has been shown to degrade performance. LPT (Liao et al., 2025a) introduces long-form CoTs distilled from large reasoning models, such as R1 (DeepSeek-AI et al., 2025), with explicit emphasis on expanding thoughts and encouraging models to deliberate more thoroughly across perception-heavy tasks. Our work builds on this line: rather than restricting CoTs to fixed stages or linear narratives or reusing existing MCQs, we synthesize diverse questions and complex reasoning structures, explicitly scaling along three axes: scale, complexity, and reasoning depth.

## 6 CONCLUSION

We introduced a scalable framework for synthesizing vision-centric reasoning data that unifies scale, complexity, and cognitive richness, producing 1M+ high-quality data. Our pipeline leverages grounded metadata and composition hardening to create diverse, verifiable tasks that go well beyond prior math-centric efforts. By coupling VLMs with reasoning LLMs, we show how frontier models can be "given eyes" to distill rich cognitive behaviors while maintaining scalability. Our experiments demonstrate that this data not only advances open VLMs on vision-centric tasks, but also transfers across modalities and out of distribution data.

## 7 LIMITATIONS

While our models show consistent gains across image MCQ benchmarks, several limitations remain. First, our work is primarily experimentally driven. We demonstrate what works at scale but an explicit theoretical framework is deferred to future works. Second, we only synthesize MCQ problems. Third, despite filtering, some synthesized MCQs remain vague or ambiguous; Finally, our image pool is restricted to DOCCI, and our approach relies on high-quality captions.

## REPRODUCIBILITY STATEMENT

To ensure the reproducibility of our work, we provide comprehensive details of our proposed model, including model specifications, data generation pipeline, pre-processing steps, training setups in the main paper and supplementary materials, along with the specific hyperparameter configurations and the prompts for data generation. The artifacts of this work such as datasets and models are planned to be publicly released upon publication, to facilitate future works in visual reasoning.

## ETHICS STATEMENT

This work adheres to the ICLR Code of Ethics. We use the public DOCCI dataset to generate a large-scale synthetic visual reasoning dataset, an approach that avoids the privacy and consent issues inherent in new real-world data collection. No new human subjects were involved. We acknowledge that the large models used for synthesis may introduce societal biases into the final dataset, and we recommend a full bias audit as future work. To advance open-source research and promote transparency, we will release our dataset and models. We advocate for the responsible application of this technology in a long-term development.

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

# APPENDIX

## A  DETAILS ON DATA GENERATION STAGES AND PROMPTS

### A.1  STAGE 1: OBJECT-CENTRIC PROBLEM GENERATION

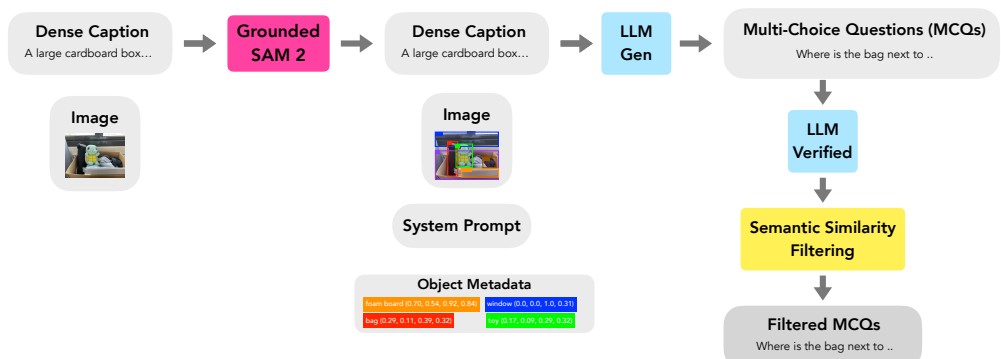

Figure 6: **Stage 1 Generation Pipeline.**

**Object-oriented metadata generation.** Our data generation pipeline is composed of two primary stages: a scalable data generation stage focused on quantity and a compositional hardening stage focused on complexity. The first stage is illustrated in Figure 6. Our pipeline begins by leveraging a dense captioning model to produce a detailed textual description of the image. Concurrently, we process the image with Grounded SAM-2 (Ren et al., 2024) to extract a rich set of open-vocabulary bounding boxes and corresponding object tags, which we term Object Metadata. This metadata, including labels and coordinates (e.g., bag $(0.29, 0.11, 0.39, 0.32)$), is integrated with the dense caption into a structured system prompt. This prompt guides a large language model (LLM Gen) to create object-centric MCQs that are inherently focused on specific visual elements. On average, each DOCCI image contains a median of 10.7 bounding tags, determined after applying a confidence cutoff of 0.9 from the detection source variable of the Florence-2 feature extractor (Xiao et al., 2024), a 0.7B vision encoder–decoder model, within the Grounded SAM-2 bounding box generation pipeline. To stabilize the bounding box–grounded MCQ generation process in the Stage 1 pipeline over diverse object-context, we constrain the maximum number of same-category instances per image to 9 (e.g., up to nine "tree" bounding box tags in a visual scene image).

**Semantic filtering for question diversity.** To ensure the quality and diversity of the generated data, we employ a two-step filtering process. First, an Qwen3-30B-A3B-Instruct-2507 (Yang et al., 2025) based **LLM verifier** assesses the factual correctness and logical soundness of each generated MCQ. Second, a semantic similarity filtering step is applied to discard redundant or overly similar questions, thereby enhancing the diversity of the final dataset.

For the semantic filtering, we represent each MCQ $i$ by a question stem embedding $q_i$, a selected answer embedding $a_i$, and an optional set of category tags $c_i$. Embeddings are derived from all-MiniLM-L6-v2, a 22M text embedding model in SentenceTransformer (Reimers & Gurevych, 2019), after standard text normalization. We define a composite similarity score between two MCQs, $i$ and $j$, as a weighted combination:

$$s(i, j) = \lambda_s \cos(q_i, q_j) + \lambda_a \cos(a_i, a_j) + \lambda_c \mathcal{J}(c_i, c_j), \tag{1}$$

where $\cos(\cdot, \cdot)$ denotes cosine similarity and $\mathcal{J}(\cdot, \cdot)$ is the Jaccard index for category tags. The weights $\lambda_s$, $\lambda_a$, and $\lambda_c$ balance the contribution of the question stem, answer, and category similarity, respectively.

For each newly generated MCQ $i$, we query an k-nearest neighbors index to retrieve its top-$k$ closest neighbors from the set of already accepted questions. We then compute the composite similarity $s(i, j)$ for these neighbors. If its maximum similarity to any previously accepted question $j$ exceeds the deduplication threshold $\tau_{dup}$:

$$\max_{j < i} s(i, j) \geq \tau_{dup}, \tag{2}$$

where $\tau_{dup}$ is a value set to $0.82$ after an ablation study on the filtered question quality. Otherwise, the question is added to our filtered dataset and its embedding is added to the index.

## A.2 ADDITIONAL ANALYSIS OF DATA DIVERSITY

To investigate the source of the scalability gains reported in Section A.1, we analyze the semantic diversity of the generated supervision. We hypothesize that conditioning on dense captions alone leads to *MCQ synthesis collapse*, where the generator repeatedly targets the most salient image features (*e.g.*, dominant objects, colors), creating redundant training signals. In contrast, we posit that conditioning on open-vocabulary bounding boxes (LGT) forces the generator to explore the long tail of visual concepts.

We test this by sampling $N = 1000$ questions generated from three shared seed images for both LGT (Ours) and LPT (Baseline). We encode these questions using SentenceBERT (Reimers & Gurevych, 2019)`all-MiniLM-L6-v2` and estimate their density in the semantic space using Gaussian Kernel Density Estimation (KDE).

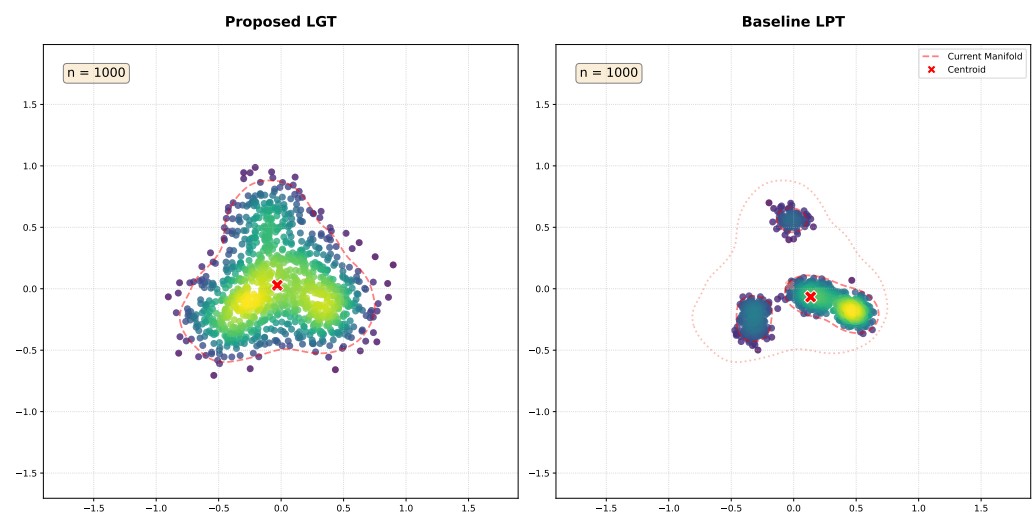

Figure 7: **Semantic Manifold Comparison.** KDE contours (95% confidence) reveal that caption-only generation (LPT, right) suffers from mode collapse, clustering tightly around coarse visual features. Object-grounded generation (LGT, left) prevents this collapse, expanding the supervision manifold to cover finer-grained spatial and attribute reasoning.

As shown in Figure 7, the baseline distribution exhibits severe mode collapse, characterized by high density in a narrow semantic region. Quantitatively, LGT demonstrates superior coverage:

- **Semantic Spread:** LGT yields a **3.2× wider** distribution (mean Euclidean distance to centroid) than the baseline.

- **Information Density:** LGT significantly reduces redundancy, with an average pairwise cosine similarity of **0.61** compared to **0.82** for LPT.

These results confirm that object-level metadata prevents MCQ synthesis collapse and maintains diversity at scale. By preventing saturation in the supervision signal, LGT enables the continuous performance scaling observed in our main results, whereas caption-based methods rapidly hit diminishing returns.

## A.3 VISION-CENTRIC PROMPT TEMPLATES

When the previous LLM based text-to-text MCQs generation frameworks (Liao et al., 2025a) are highly relied on the image caption quality of the text, injection visual-grounded information has not been well explored. We carefully design a coordinate information guided prompt for the vision-centric MCQ generation with one main task instruction prompt, one core principle, and four auxiliary guidelines as shown in A.3. We will open source the full data generation pipeline with the exact prompt instruction.

> **Prompt Task**
>
> You are a computer vision expert generating object-centric visual questions that require precise examination of a specific bounded region. Given a detailed image description and a target object with its bounding box coordinates, create challenging multiple-choice questions that demand careful analysis of the object within its bounded region and its contextual relationships. **Generate exactly `{{ num_questions | default(4) }}` multiple-choice questions.**

**Prompt Task** At the start of the process, we add a "computer vision expert" role-playing instruction; therefore, the text-LLM can better interpret *bounding box coordinates* and image tags while generating questions. In our experiments, using this "computer vision expert" instruction led to a noticeable improvement: the instruction-following F1 score rose from 96.3 to 99.2 during Stage 1 data generation. Similar role-playing (Zhang et al., 2025b), self-reflection (Shinn et al., 2023), and task-activating prompting (Yang et al., 2023) approaches have also been explored in recent work on domain-specific MCQ generation, such as in medical and speech understanding.

> **Core Principles**
>
> - **Object-Centric Focus**: Every question must center on the specific object within the provided bounding box.
> - **Spatial Precision**: Questions should require locating and examining the exact bounded region.
> - **Contextual Relationships**: Explore how the target object relates to its surroundings and other elements.
> - **Multi-Level Analysis**: Progress from basic object properties to complex spatial and functional relationships.

**Core Principle.** To strengthen the model's ability to reason over complex spatial and visual information in MCQs, we perform an error analysis on MTVQA (Tang et al., 2024) and MegaBench (Chen et al., 2024a) benchmarks that do not overlap with our test set, using the `QwenVL-2.5` report as reference. The analysis reveals that most failure cases stem from two sources: 55.1% misunderstanding spatial relationships (e.g., identifying where an object mentioned in the question is located in the image) and 32.2% misinterpreting contextual information (e.g., inferring the function or color of the subject). Motivated by these observations, we establish four vision-centric principles to guide question construction, leveraging dense captions, bounding box labels, and coordinate information.

**Additional Auxiliary Information.** We introduce auxiliary rules to further refine question generation in alignment with our core principles. These include guidelines on how to incorporate meta-information when framing vision-centric question categories, strategies to encourage deeper investigation of visual commonsense beyond what dense-caption-based generation typically captures, refinements to both input and output formats, and a final reminder to avoid directly disclosing bounding box coordinates in the question text. Instead, the text should be used to construct the subject that grounds the visual target.

In summary, LGT employs a carefully designed prompt instruction comprising $1,987$ tokens, which is substantially longer than the simpler vision-instruction prompt of $419$ tokens used in Liao et al. (2025a). The full prompt is provided in detail below.

> **Question Categories**
>
> Distribute your `{{ num_questions | default(4) }}` questions among the following categories:
>
> 1. **Specific Region Analysis** (`{{ ((num_questions | default(4)) * 0.25) | round | int }}` question`{{ 's' if ((num_questions | default(4)) * 0.25) | round | int != 1 else " }}`):
>    - Object attributes within the bounding box (color, texture, material, size, orientation).
>    - Object state and condition (pose, activity, physical state).
>    - Object parts and components visible within the bounded region.
>    - Visual details that distinguish this object from similar objects.
>
> 2. **Object-Environment Interactions** (`{{ ((num_questions | default(4)) * 0.25) | round | int }}` question`{{ 's' if ((num_questions | default(4)) * 0.25) | round | int != 1 else " }}`):
>    - Spatial relationships between the target object and immediate surroundings.
>    - How the object interacts with or relates to nearby elements.
>    - Environmental context affecting the object's appearance or function.
>    - Lighting, shadows, or reflections involving the target object.
>
> 3. **Comparative & Relational Questions** (`{{ ((num_questions | default(4)) * 0.25) | round | int }}` question`{{ 's' if ((num_questions | default(4)) * 0.25) | round | int != 1 else " }}`):
>    - How this object compares to other objects in the scene.
>    - Relative positioning, size, or orientation compared to other elements.
>    - Object hierarchies or groupings involving the target object.
>    - Contextual significance of the object within the overall scene.
>
> 4. **Functional & Semantic Analysis** (`{{ ((num_questions | default(4)) * 0.25) | round | int }}` question`{{ 's' if ((num_questions | default(4)) * 0.25) | round | int != 1 else " }}`):
>    - Object's purpose or function within the scene context.
>    - How the object is being used or what role it plays.
>    - Semantic relationships between the object and scene narrative.
>    - Implied actions or processes involving the target object.

> **Design Guidelines**
>
> - **Implicit Object Targeting**: Questions should focus on the target object without explicitly revealing bounding box coordinates in the question text.
> - **Object Identification Challenge**: Questions must require the reader to first identify and locate the target object based on contextual clues and object description.
> - **Progressive Complexity**: Start with direct object attributes, then move to spatial relationships, then to complex contextual analysis.
> - **Precise Language**: Use specific spatial terms (e.g., *adjacent to, overlapping with, positioned above*) and descriptive object references.
> - **Distractors Strategy**: Create plausible wrong answers that might apply to other objects in the scene but not the target object.
> - **Coordinate Disclosure**: **DO NOT** mention bounding box coordinates in the question text.

- **Design for Multiple-Choice**: Provide 4 answer options (A, B, C, D) with one correct answer and three plausible distractors that require careful inspection to rule out.
- **Clarity, Specificity, and Brevity**: Formulate clear, focused questions that are detailed enough to challenge the reader, avoiding ambiguity or reliance on general knowledge.

### Input & Output Format

**Image Description:** `{{ image_description }}`
**Target Object Analysis:**

- **Object**: `{{ bbox_label }}`
- **Image Dimensions**: `{{ image_width }}` × `{{ image_height }}` pixels

**Structured Output Example:**

```
1. <question> Your question here </question>
   <choices> (A)... (B)... (C)... (D)... </choices>
   <answer> object_label, [x1, y1, x2, y2], actual_answer
   </answer>
   <type> reasoning type here </type>
```

### ⚠ Critical Reminders

- ● You must generate **exactly `{{ num_questions | default(4) }}`** questions.
- ✕ Questions **MUST NOT** disclose bounding box coordinates or specific object labels. Use generic terms like "the object," "the item," or "the element" instead of "`{{ bbox_label }}`".
- ● Answers **MUST INCLUDE** the exact object label and integer coordinates in the specified format: "`{{ bbox_label }}, [{{ bbox_coordinates[0]|round|int }}, {{ bbox_coordinates[1]|round|int }}, ...], [specific answer]`".
- ◎ Questions must focus on visual properties or relationships of the object within the specified bounded region, requiring careful inspection.

### A.4 STAGE 2: COMPOSITIONAL PROBLEM GENERATION

While Stage 1 generates a large and diverse set of grounded questions, a significant portion of them are relatively simple and can be solved directly by the base VLM. To push the model's reasoning capabilities further, Stage 2 introduces a **composition hardening** algorithm designed to systematically increase problem complexity.

The approach is straightforward yet effective: for a given image, we randomly sample up to five question-answer pairs generated in Stage 1. These simpler problems, along with the global image caption, are provided as input to a generator LLM. The LLM is then tasked with composing these individual questions into a single, more challenging multi-hop problem that requires higher-order reasoning to solve. Following generation, we apply a similar filtering protocol as in Stage 1: the generator model is prompted to solve its own composed question, and we retain only those problems with a high answer consistency threshold ($\tau \geq 0.8$), ensuring both difficulty and verifiability. The specific prompt used for this stage is detailed in Appendix A.4.



**Prompt Task**

You will be given a description of an image and up to 5 different easy problems asking about the image. Use the questions to create a single, creative and hard question to solve.

- The question should be in a multiple-choice format with 4 options, just like the given questions.
- The composed question should be much harder than each of the individual sub-questions provided.
- You should focus on the perceptual capabilities (e.g. counting objects, detecting color or texture of an object, relative location of the objects, the angle of the image, detecting letters, etc) and creatively use them to make a harder question.
- Do not simply ask about an enumeration of these features, and you may focus on one specific aspect among them. But make sure that the new question is harder to solve.
- Only use english in your problem.

Your output should be exactly formatted as:
```
Hard problem
your hard question
(A) your hard problem option A
(B) your hard problem option B
(C) your hard problem option C
(D) your hard problem option D
Correct answer:  your correct answer
```



**Simple CoT VLM distillation and Thought Expansion** A core challenge in synthesizing reasoning traces is that open-weight VLMs often lack the rich, non-linear cognitive behaviors (e.g., subgoal setting, backtracking) seen in frontier LLMs, while human annotation is prohibitively expensive at scale. To address this, we adopt the two-step distillation process from LPT, as illustrated in Figure 1 (bottom). The prompt templates used for this distillation process are adapted from a baseline in Liao et al. (2025a).

First, to ensure the reasoning remains in-distribution for the target model, we prompt a VLM ($\mathcal{M}_{VLM}$) with the image and MCQ to produce a concise initial rationale, termed a "Simple CoT". Naively sampling from a powerful reasoning LLM directly often yields out-of-distribution traces that degrade downstream performance. Second, we perform a **thought expansion** step, where the Simple CoT is used to prime a stronger reasoning LLM ($\mathcal{M}_{Reason}$), which expands upon the visually-grounded trace while injecting more complex problem-solving strategies.

Crucially, our approach scales the model choice with task complexity. For the ~750K simpler problems in Stage 1, we use efficient models (`Qwen2.5-VL-7B` and a 32B-scale $\mathcal{M}_{Reason}$). For the more complex compositional problems in Stage 2, we leverage frontier models like `Qwen2.5-VL-72B` and R1 to generate the richest possible reasoning chains.

A.5 LOCAL QWEN COT VERIFICATION PROMPT

A critical component of our data synthesis pipeline is ensuring the logical fidelity of the expanded reasoning traces. While the "thought expansion" step (Section 2.3) enriches Simple CoTs with complex cognitive behaviors, the powerful reasoning LLM can sometimes produce plausible-sounding but factually incorrect reasoning paths that deviate from the ground-truth answer. To filter these inconsistencies at scale without relying on expensive external judges, we designed the following verification prompt. This prompt tasks a smaller, local model (`Qwen-32B-A3B-2507-Instruct`) to act as an efficient verifier. The model must assess whether the generated reasoning trace (`Reflection`) logically supports the known correct `Answer`, effectively serving as a high-throughput quality control mechanism for our SFT and preference data.

---

**Prompt Task**

You will be given a visual question, its answer, and a reflection on an initial thought process. The provided answer is always correct, but the reflection may sometimes be inconsistent with this answer. Your task is to check if the reflection logically and factually supports the provided answer.

---

**Verification Process**

You will check for consistency by following these steps:

1. **Understand the Question and the Answer:** First, fully comprehend what is being asked and what the correct final answer is.

2. **Derive the Answer Solely from the Reflection:** Carefully read the `Reflection` text and determine what conclusion it leads to, ignoring the provided `Answer`.

3. **Check for Consistency:** Compare the answer derived from the `Reflection` (Step 2) with the provided ground-truth `Answer`.

- - - - - - - - - - - - - - - - - - - - - - - - - - - - - - - - - - - - - - - -

**Output Requirement:** At the end of your reasoning, you must answer with $\boxed{Yes}$ if the Reflection is consistent with the Answer; otherwise, answer $\boxed{No}$.

---

```
Input Format
```

---

**# Question:** `<question_text>`
**Answer:** `<answer_text>`

**# Reflection on the initial thought**
**Reflection: ...** `<last_30_words_of_reflection>`

---

## A.6 ADDITIONAL EXPERIMENTAL RESULTS

### A.6.1 MULTIMODAL AUDIO UNDERSTANDING AND OMNI MODEL

We emphasize our data is completely vision-centric. Here, we conduct experiments on out-of-domain (OOD) complex audio reasoning tasks using the Qwen2.5-Omni-7B-Instruct (Xu et al., 2025) model on the Multimodal Audio Understanding (MMAU) benchmark (Sakshi et al., 2024). Our methodology leverages the modular architecture of Qwen-Omni, which separates the reasoning component ("thinker") from the answer generation component ("talker"). We first fine-tune only the "thinker" dense module using our synthesized data. We keep our modality-specific audio encoder frozen when tuning the backbone dense model. After this stage, the fine-tuned "thinker" is merged back with the original, pre-trained "talker" module. This strategy aims to enhance the model's intrinsic reasoning capabilities without directly altering the audio-specific knowledge contained within the "talker" module, thereby mitigating catastrophic forgetting.

The results, presented in Table 6, demonstrate a surprising and significant positive transfer. The baseline Qwen2.5-Omni-7B-Instruct already establishes a strong performance with an MMAU average of 71.00, surpassing proprietary models like Gemini-2-Flash (67.03). However, not all reasoning data transfers effectively; fine-tuning on Virgo,a visual math dataset, leads to a substantial performance degradation to 62.60, indicating negative transfer. In contrast, our LGT data shows positive scaling behaviors.

Training on 500k LGTs already pushes the model's performance to 71.80, outperforming the other leading omni-models. By scaling to 1M LGTs, our model achieves a new state-of-the-art average score of 72.32 among the tested configurations. These gains are driven by significant improvements in non-speech acoustic reasoning, with the **MMAU-Sound** score rising from 76.77 to 78.30 and the

Table 6: Omni's generalization Results benefited from Long Grounded Thoughts (LGT) on audio understanding. Best scores in each column are in bold.

| Approach | MMAU-Sound | MMAU-Music | MMAU-Speech | MMAU-avg |
|---|---|---|---|---|
| Gemini-2-Flash | 68.93 | 59.37 | 72.87 | 67.03 |
| GPT-4o | 63.24 | 49.93 | 69.33 | 60.80 |
| Qwen2.5-Omni-7B-Instruct | 76.77 | 67.33 | **68.90** | 71.00 |
| + Virgo | 64.20 | 59.30 | 64.30 | 62.60 |
| + LPT | 76.63 | 67.56 | 66.93 | 70.23 |
| Ours | | | | |
| + 130k | 76.60 | 67.80 | 67.40 | 70.23 |
| + 250k | 76.90 | 68.90 | 67.80 | 70.23 |
| + 500k | 77.20 | 69.50 | 68.82 | 71.80 |
| + 1M | **78.30** | **70.20** | 68.80 | **72.32** |

**MMAU-Music** score increasing from 67.33 to 70.20. This result underscores that enhancing core reasoning abilities with our vision-centric data can positively transfer to and improve performance on OOD audio tasks. This complex tracing of learning benefits also extends to more challenging omni-modal tasks involving dual-modal inputs (e.g., vision and audio). Our experimental results with long reasoning traces further support the recent finding (Rouditchenko et al., 2025) that it is possible to enhance audio content reasoning ability solely by injecting high quality reasoning text data. We provide a qualitative example in 9 on how reasoning traces on recognizing complex and compositional sound and speech events.

Table 7: MMLU-Pro Results

| Model | Acc |
|---|---|
| Qwen2.5-VL-7B-Instruct | 47.15 |
| VLAA-thinking | 21.56 |
| Virgo | 37.95 |
| Ours (LPT) | 50.77 |
| Ours (SFT Only) | 50.13 |
| Ours (SFT + DPO) | 47.07 |
| Ours (SFT + GRPO) | 47.00 |

### A.6.2 TEXT-ONLY MMLU-PRO

MMLU-Pro (Wang et al., 2024) extends the original MMLU (Hendrycks et al., 2021) by incorporating more reasoning-intensive questions and enlarging the answer choice set. It covers 14 broad domains—including mathematics, physics, chemistry, and others—and consists of over 12,000 questions in total. In table 7 we show the extended table that was presented shorted in the main work.

### A.6.3 EMBODIED OPEN-ENDED QA BENCHMARK

We evaluate on a modified version of *Needle in the Embodied Haystack* (NiEH) (Kim & Ammanabrolu, 2025), focusing on the **single-evidence** setting where one frame in a time-ordered trajectory suffices to answer the question. The test set contains 829 image–question pairs. In our evaluation, we restrict the input to a 2048-token window centered on the ground-truth (answer-bearing) frame, preserving temporal order so that multiple adjacent frames remain visible around the evidence.

**Prompting.** For the Qwen-7B-VL baseline, we use the original paper's prompt. For our models, we apply a fixed system prompt and the following task instruction: *"These images are the agent's view in time order. Answer the question given the images. Do not include explanation or reasoning in the answer tags. Answer with a single word or words."*

Table 8 summarizes results on the modified NiEH single-evidence benchmark (EM).

Table 8: Results on the modified NiEH single-evidence benchmark. Higher is better. Exact Match.

| Model | Score |
|---|---|
| Qwen2.5-VL-7B-Instruct | 47.55 |
| + VLAA-thinking | 46.75 |
| + LPT | 51.95 |
| + Ours (SFT Only) | 48.24 |
| + Ours (SFT + GRPO) | 39.10 |
| + Ours (SFT + DPO) | **56.34** |

Table 9: SFT and RL ablations across data generation algorithms

| Data Generation Algo. | Starting Point | SFT Data | RL Algo. | RL Data | Avg Perf. |
|---|---|---|---|---|---|
| LPT | BaseModel | 750K | None | None | 0.682 |
| Ours | BaseModel | 750K | None | None | **0.716** |
| LPT | BaseModel | 0 | GRPO | 1K | 0.673 |
| | | | | 5K | 0.682 |
| | | | | 10K | **0.695** |
| | | | | 50K | 0.683 |
| | | | | 100K | 0.669 |
| LPT | SFT | 750K | GRPO | 1K | 0.678 |
| | | | | 5K | 0.658 |
| | | | | 10K | 0.662 |
| | | | | 50K | 0.685 |
| | | | | 100K | **0.709** |
| Ours | BaseModel | 0 | GRPO | 70K | 0.704 |
| | SFT | 750K | DPO | 70K | 0.737 |
| | SFT | 750K | DPO | 129K | 0.740 |
| | SFT | 750K | GRPO | 70K | **0.757** |
| | SFT | 750K | GRPO | 129K | 0.746 |

Table 10: Detailed cognitive behaviour statistics for various datasets. Values represent the average count of the behaviour on a subsample of 1000 examples of the dataset. Quantification methodology and terminology follows from Gandhi et al. (2025); Liao et al. (2025a).

| Behaviour | LongPerceptual Thoughts | LongGrounded Thoughts (s1) | LongGrounded Thoughts (s2) | allenai/pixmo-ask-model | DriveLMMo1 | LENS | SCI-Reason | VLLA-Thinking | Virgo |
|---|---|---|---|---|---|---|---|---|---|
| Subgoal setting | 0.036 | 0.12 | 0.55 | 0.017 | 0.235 | 0 | 0.032 | 0.17 | 0.24 |
| Backtracking | 0.354 | 0.35 | 0.68 | 0 | 0 | 0 | 0 | 0.37 | 0.79 |
| Verification | 0.26 | 0.33 | 0.76 | 0 | 0 | 0 | 0 | 0.45 | 0.89 |

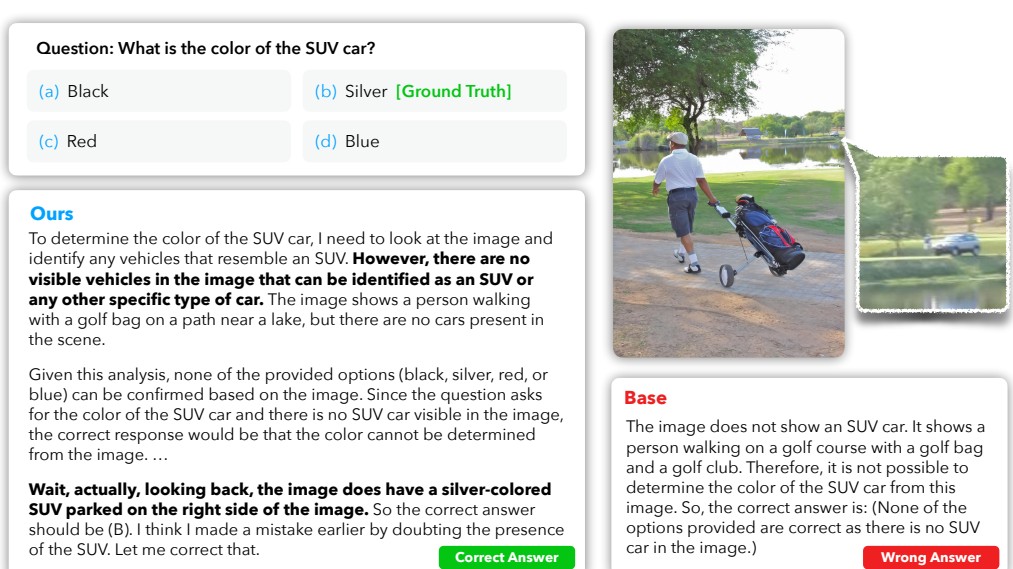

Figure 8: Additional qualitative example of a reasoning trace from the model post-trained on our data vs the base model.

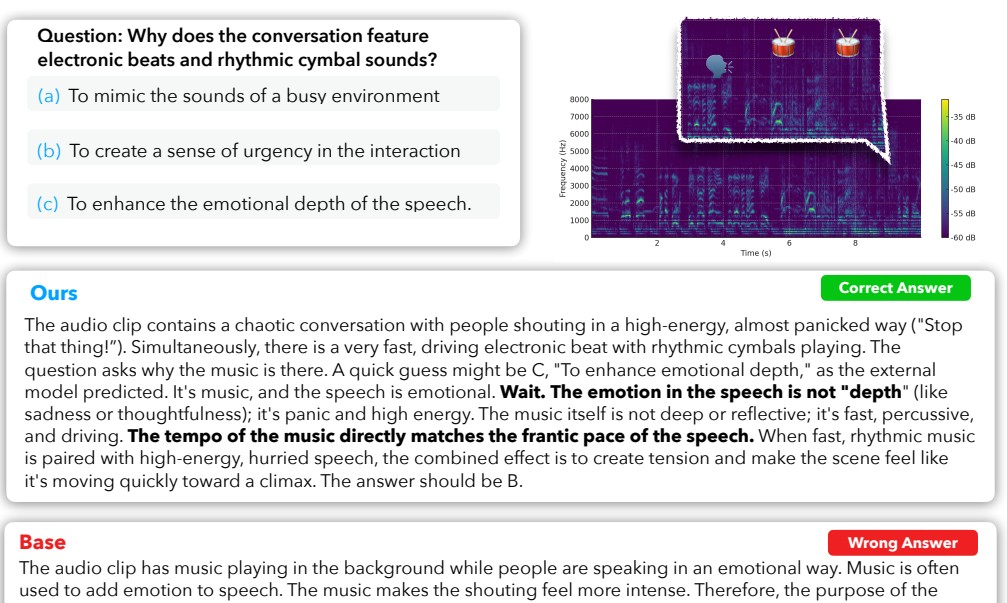

Figure 9: Temporal reasoning improvement illustrated by a qualitative example of a reasoning trace from the Qwen-2.5 Omni model post-trained on our data, compared to the base Qwen-2.5 Omni model, on an unseen audio reasoning question involving joint speaking and sound events.

