# OpenReview forum: "Long Grounded Thoughts: Distilling Compositional Visual Reasoning Chains at Scale"
_ICLR.cc/2026/Conference — Submitted to ICLR 2026_

### Official Review · Reviewer_8Kih · 2025-10-20

**Soundness:** 2
**Presentation:** 2
**Contribution:** 1
**Rating:** 2
**Confidence:** 2

**Summary:**

The paper introduces Long Grounded Thoughts, a data generation framework designed to produce vision-centric questions across varying difficulty levels. The framework operates in two stages: (1) a scaling stage, where bounding boxes and tags are used to generate grounded VQA questions, and (2) a complexity stage, where these questions are composed into higher-level reasoning tasks. The method leverages a Vision-Language Model (VLM) to generate chain-of-thought (CoT) traces and a Large Language Model (LLM) to perform the reasoning. Experimental results on multiple benchmarks show that models fine-tuned on the generated data outperform baseline models.

**Strengths:**

•	The analysis of post-training in VLMs was comprehensively conducted.

**Weaknesses:**

•	The novelty of the study is limited. Many parts of the study are adopted from LongPerceptualThought (LPT) method which makes it an extended study. The authors include bounding box and tags in addition to the description of the image and increase the complexity of the generated questions. However, the main configuration relies on the previous LPT study.

•	The process of reasoning is not explained and depicted clearly. VLM generates CoT with predicted answers. Dense caption is also provided for the reasoning step according to Figure 1. If we use the CoT process, then why do we need the dense caption to answer the question? How did these two affect the LLM reasoning process? Does providing predicted VLM answer create any bias for LLM reasoning step? How did models generate different answers than VLM’s predicted answer? Does the model answer the question rely on dense caption only?

•	The method section is not explained clearly, instead it gives comparison to the LongPerceptualThought (LPT) and evaluation results.

**Questions:**

•	What are the contributions of the study?

•	The authors mentioned that they teach basic cognitive behaviors: verification, backtracking and correction. However, they did not explain how the process works.

•	Table 1 compares the datasets according to their size, domain and data type. However, the study focuses on complexity and richness, besides scale. How are other datasets compared on these terms?

•	The reasoning traces are distilled from the models, but it is unclear how reliable or trustworthy these traces are. Could the authors conduct a small-scale analysis or evaluation to assess the consistency and accuracy of the generated traces?

•	Could the authors clarify why the performance of the online RL appears to plateau, and what might have occurred around the 70K example mark to cause this shift?

•	The study follows LongPerceptualThought (LPT) for many steps. In what ways does this study differ from it?

•	The reviewer does not understand the connection between diversity, scalability of LPT with Figure 2? The figure shows average performance based on dataset size. How to infer diversity in this figure?

•	The references for the Appendix are not stated well (see supp material).

•	Figure 3 shows distribution of correct rollouts by data split. Do the LPTv1 and others obtain the same size of data in the figure?

•	Figure 4 shows the cognitive behaviors in CoT. How did you analyze and evaluate the results in Figure 4?

---

> ### Author Response · Authors · 2025-11-21
> **Response to Reviewer 8Kih - Part 1**
>
> We sincerely appreciate the time and effort you invested in reviewing our work. We recognize that some aspects of our presentation may not have been as clear as they could be, and we're grateful for the opportunity to clarify these points. We've carefully considered each concern and made substantial revisions to address them. We hope the clarifications below, along with the new experimental results we've added, will help demonstrate the distinct contributions of our work.
>
> ----
>
> ### Concern 1: Novelty and Contributions
>
> We appreciate this concern and recognize we could have better distinguished our contributions from LPT in the original submission. While we build on LPT's thought-expansion mechanism, we solve fundamentally different problems:
>
> - **LPT's focus:** Expanding reasoning traces for existing questions
> - **Our focus:** Generating questions at scale + composing them into harder problems + distilling richer reasoning traces with more cognitive behaviours.
>
> Specifically, LPT saturates at ~100K examples due to MCQ synthesis  collapse when generating from captions alone. Figure 2 in our paper shows their curve plateaus while ours continues scaling to 1M+. Our key insight is that object-level grounding (bounding boxes + tags) prevents this collapse. To address your concern about demonstrating diversity more concretely, **we've now added Section A.2 in the appendix with quantitative diversity analysis**. Using SentenceBERT embeddings on N=1000 questions from shared seed images, we show **LGT achieves 3.2x wider semantic distribution and significantly lower redundancy (cosine similarity 0.61 vs 0.82 for LPT)**.
>
> The compositional hardening algorithm (Stage 2) is also entirely novel—systematically merging simple questions into verifiable multi-hop problems. This is evidenced by the dramatic increase in subgoal setting (0.12 → 0.55) as well as as other cognitive behavior. We have made this clear in Table 10 in the supplementary material.
>
> ### Key Contributions:
>
> 1. **Grounded Synthesis for Scale:** First demonstration that object-level metadata prevents mode collapse, enabling scaling from 100K → 1M+ examples (Section 2.1, Appendix A.2)
> 2. **Compositional Hardening for Complexity:** Novel algorithm for systematically increasing problem difficulty through question composition (Section 2.2, Table 10)
> 3. **1M+ Open Vision-Reasoning Dataset:**  Open dataset with structured cognitive behaviors (subgoal setting, backtracking, verification), supporting full post-training pipeline (SFT + RL)
>
> ### Empirical Contributions:
>
> 4. **Comprehensive Post-Training Analysis:** Comprehensive systematic study of SFT vs. offline RL vs. online RL for VLMs with high-quality reasoning data (Section 4.2)
> 5. **Cross-Modal Transfer:** Demonstration that vision reasoning transfers to text (MMLU-Pro: +2.98%) and audio (MMAU: +1.32%) (Section 4.3)
>
> We've clarified these distinctions in the revised introduction and explicitly added to Table 1 that we are the first open dataset at 1M+ scale with full post-training pipeline support.
>
> ---
>
> ### Misunderstanding 1: Reasoning Process Clarity
>
> Thank you for this question. We realize our Figure 1 may have created confusion about when different components are used. Let us clarify the distinction between data generation and model usage:
>
> ### During Data Generation (what Figure 1 shows):
>
> 1. VLM sees: Image + MCQ → outputs Simple CoT (z₁) + answer (a₁)
> 2. Reasoning LLM sees: Dense caption + MCQ + Simple CoT (z₁) → outputs expanded Long CoT (z₂)
>
> The ***caption is used here because the reasoning LLM cannot see images***. It indirectly grounds the LLM's expansion in visual content.
>
> ### During Model Training & Inference (what users see):
>
> - **Input:** Image + MCQ only
> - **Output:** Long CoT + answer
> - **No caption involved**
>
> **The caption is never seen by the trained model**. It exists only during the synthesis phase.
>
> We chose this two-stage approach specifically to keep reasoning traces in the VLM's output distribution. As stated in Section 2.3: "Sampling from a VLM ensures that the synthesized CoTs remain within distribution... This preconditioning allows the reasoning LLM to expand familiar traces while injecting richer, non-linear problem-solving strategies."
>
> ---
>
> ### Addressing Your Specific Questions:
>
> --
>
> **"Does providing predicted VLM answer create bias?"**
> Yes, intentionally. We want CoTs in the VLM's distribution, not the LLM's. Similar to Liao et al., we found this fundamentally important. This is mentioned in Section 2.3.
>
> --
>
> **"How did models generate different answers?"**
> When the VLM is wrong (a₁⁻), we create preference pairs for DPO (Section 3).
>
> --
>
> **"Does model rely on caption only?"**
> No. The caption is never seen by the trained model during training or inference.
> We've added explicit clarification of this distinction in the revised Section 2.3 to prevent future confusion.

---

> > ### Author Response · Authors · 2025-11-21
> > **Response to Reviewer 8Kih - Part 2**
> >
> > ### Misunderstanding 2: Method Section Clarity
> >
> > We appreciate your feedback on the method section presentation. We want to respectfully note that none of the other reviewers raised concerns about methodology clarity. Sections 2.1, 2.2, and 2.3 contain detailed algorithmic descriptions with formal notation.
> >
> > Section 2 (pages 2-5) contains:
> > - **Section 2.1:** Object-centric generation algorithm with formal notation (page 3)
> > - **Section 2.2:** Composition hardening algorithm with formal notation (pages 4-5)
> > - **Section 2.3:** Two-stage distillation process with formal notation (page 5)
> >
> > Each section includes algorithm descriptions, choices, and justifications. Comparisons to LPT appear only where relevant to highlight our innovations. We believe this structuring makes our methodological innovations clearer while still providing necessary context.
> >
> > However, we recognize that our comparisons to LPT throughout may have made the novel components less prominent to you specifically. We have emphized them with more clear wording in the editted manuscript.
> >
> > ---
> >
> > ### Question: Figure 2 and Diversity
> >
> > Thank you for pressing on this point. Figure 2 measures the **consequence** of diversity loss on model performance. When question diversity decreases, models see repetitive patterns, and performance saturates. Intuitively:
> >
> > - LPT's curve plateaus at 100K → indicating saturation from repetitive questions
> > - Our curve continues improving to 1M → suggesting sustained diversity
> >
> > Following your recommendation, we've conducted **a new diversity analysis on the MCQ synthesized by our approach vs LPT**, which we've added to the supplementary material (Section A.2). Specifically, we sampled N=1000 MCQ questions generated from three shared seed images for both LGT (Ours) and LPT (Baseline). We encoded these questions using SentenceBERT and estimated their semantic spread and average pairwise cosine similarity in the embedded space.
> >
> > ### Results:
> >
> > - **Spread:** LGT yields a 3.2x wider distribution (mean Euclidean distance to centroid) than the baseline
> > - **Information Density:** LGT significantly reduces redundancy, with an average pairwise cosine similarity of 0.61 compared to 0.82 for LPT
> >
> > This quantitatively reinforces that our grounding-based approach maintains substantially higher diversity at scale. We've included additional visualizations in the appendix to illustrate this difference.
> >
> > ---
> >
> > ### Question: Cognitive Behaviors (Verification, Backtracking, Correction)
> >
> > These behaviors emerge naturally from our two-stage distillation process described in Section 2.3. This is the beauty of the approach! In Stage 1 (Simple CoT), the VLM generates an initial reasoning process that is typically linear—the VLM observes the image and begins reasoning, which is captured directly in z₁.   In Stage 2 (Thought Expansion), the reasoning LLM enriches this reasoning with additional structure. The LLM is prompted to "continue the thought" from z₁ and is explicitly encouraged to verify claims and consider alternatives. Notably, this is a reasoning LLM that has already internalized these cognitive behaviors so most of them arise “for free” during expansion. This is a strongly desired behaviour that avoids hand-designing them. We further employ regex-guided generation to prevent problematic patterns, as detailed in Section 2.3.
> >
> > **Our evidence for this claim is presented in Table 10**, which demonstrates that our dataset exhibits higher frequencies of verification, backtracking, and correction behaviors compared to LPT and other datasets. The quantification methodology follows Gandhi et al. (2025) for detecting these reasoning patterns, which is properly cited in our paper.
> >
> > **This entire process is thoroughly explained in Section 2.3, with detailed analysis provided in Section 4 and empirical evidence shown in Figure 4b and Table 10.**
> >
> > ---
> >
> > ### Question: Reliability of Reasoning Traces
> >
> > Excellent question. We have comprehensive automatic validation that distinguishes our approach from most synthetic datasets. Our validation employs a multi-stage pipeline to ensure trace quality:
> >
> > 1. **LLM Verifier:** Checks for logical consistency (detailed in Appendix A.4)
> > 2. **Answer Validation:** Ensures the expansion ends up with the correct answer
> > 3. **Consistency Verification:** Confirms reasoning properly supports the final answer (specific prompts in Appendix A.4)
> > 4. **Self-Verification (Stage 2):** Generator model solves its own composed questions with τ ≥ 0.8 threshold
> >
> > Beyond these direct validation methods, we observe **indirect validation** through strong empirical results across diverse benchmarks. Our approach outperforms all open-data baselines, demonstrates successful transfer to out-of-domain tasks including math and embodied QA, and even transfers cross-modally to text and audio tasks. These broad empirical successes provide compelling evidence for the reliability of our reasoning traces.

---

> > > ### Author Response · Authors · 2025-11-21
> > > **Response to Reviewer 8Kih - Part 3**
> > >
> > > ### Question: Online RL Plateau at 70K
> > >
> > > This is indeed a key finding of our work (Section 4.2, Table 3). GRPO shows fast initial gains but plateaus around 70K examples. There could be several explanations for this. For instance:
> > > 1. **Algorithmic limitation**: There might be a need for novel algorithm innovations to push beyond this scale.
> > > 2. **Data Itself**: There still might be limitations in how the data is synthesized to keep scaling further.
> > >
> > > We explicitly discuss this in Section 4.2: "GRPO yields fast gains but saturates; scaling past ∼70K does not help."
> > >
> > > ---
> > >
> > > ### Question: Differences from LPT
> > >
> > > We've addressed this above in our response to Concern 1, but to summarize the key distinctions:
> > >
> > > **Stage 1 (Scale):**
> > > - LPT: Caption-only generation, saturates at ~100K
> > > - Ours: Object-level metadata (bounding boxes + tags) enables 1M+ scaling
> > > - Innovation: First work showing spatial grounding prevents saturation at scale (Figure 2, Appendix A.2)
> > >
> > > **Stage 2 (Complexity):**
> > > - LPT: Synthesize simple MCQs, focuses on elongating reasoning
> > > - Ours: Algorithmically composes simple questions into harder multi-stage problems
> > > - Innovation: Complexity and reasoning trace richness increases  (e.g., subgoal setting: 0.12 → 0.55, see Table 10 for additional behaviours )
> > >
> > > **Overall:**
> > > - LPT: 30K examples
> > > - Ours: 1M+ examples with both SFT and RL(online, offline) data
> > > - Innovation: One of the largest vision-reasoning dataset at this scale with structured cognitive behaviors at scale.
> > >
> > > We've made this more explicit in the revised manuscript, particularly in Table 1.
> > >
> > > ---
> > >
> > > ### Question: Figure 3 - Data Split Sizes
> > >
> > > Yes, all datasets use the same data size in Figure 3. All splits shown employ the same evaluation protocol, which consists of multiple rollouts with Qwen2.5-VL. We've added explicit mention of this in the revised paper to avoid confusion.
> > >
> > > ---
> > >
> > >
> > > ### Question: Figure 4(b) Analysis
> > >
> > > We adopt the methodology from Gandhi et al. (2025) to detect cognitive patterns in reasoning traces, as cited in our paper. The computation is straightforward: an LLM determines the behaviors (using the prompts from Gandhi et al.) and we count the frequency of these patterns across 1,000 samples.
> > >
> > > Higher frequencies indicate more deliberate reasoning processes, suggesting that the model is engaging in more structured problem-solving rather than direct answer generation. This analysis demonstrates that our data contains substantially richer cognitive behaviors than baseline datasets.
> > >
> > > ---
> > >
> > > ### Minor Point: Appendix References
> > >
> > > Thank you for noting this. We will ensure all appendix references are properly formatted in the final version.

---

> > > > ### Author Response · Authors · 2025-11-21
> > > > **Response to Reviewer 8Kih  - Conclusion**
> > > >
> > > > ### Conclusion
> > > >
> > > > We hope these clarifications address your concerns. We want to emphasize the key improvements we've made in response to your feedback:
> > > >
> > > > 1. **Added diversity analysis**  (Appendix A.2) supporting Figure 2  with quantitative metrics showing direct diversity.
> > > > 2. **Improved method section** added clarifications and made wording more explicit.
> > > > 3. **Made explicit the distinction** between data generation and model inference in the figure captions.
> > > > 4. **Clarified when and why** captions are used (only during synthesis, never during training/inference)
> > > > 5. **Strengthened our contribution statements** throughout the paper
> > > >
> > > > **We respectfully note that the other reviewers understood our methodology clearly, which suggests the core content is sound**. However, we genuinely appreciate you identifying areas where additional clarity was beneficial. The revisions we've made in response to your feedback have strengthened the paper for all readers.
> > > >
> > > > We are committed to making this work as clear and accessible as possible, and **we respectfully ask the reviewer to reconsider the rating in light of these changes**. If any points remain unclear after reviewing the revised manuscript and these responses, we're happy to provide further clarification.

---

> > > > > ### Comment · Reviewer_8Kih · 2025-11-27
> > > > > **Response to Authors**
> > > > >
> > > > > Thank you for the clarifications. The authors have addressed many of my concerns, and I appreciate the additional analyses. However, the question of how other datasets compare in terms of complexity and richness, beyond size, domain, and data type, as shown in Table 1, remains unanswered.
> > > > >
> > > > > While the revisions have substantially strengthened the study and improved its quality, I still find the overall novelty to be limited. For this reason, I am raising my score to 4, but not higher.

---

> > > > > > ### Author Response · Authors · 2025-12-01
> > > > > > **Additional Response to Reviewer 8Kih**
> > > > > >
> > > > > > We thank the reviewer for raising the score and for the opportunity to clarify the remaining concerns.
> > > > > >
> > > > > > The reviewer mentions: *“how other datasets compare in terms of complexity and richness, beyond size, domain, and data type, as shown in Table 1, remains unanswered.”*
> > > > > >
> > > > > > Below, we **explicitly quantify** the **Complexity** and **Richness** of our data compared to the SoTA baseline (LPT) . We emphasize this was already addressed in our updated manuscript and previous response (Figure 4a and Table 10).
> > > > > >
> > > > > > ---
> > > > > >
> > > > > > ### 1. Dataset Complexity Comparison
> > > > > > We measure complexity by the **Pass Rate** (Consistency) over 8 attempts (as mentioned in the manuscript).
> > > > > >
> > > > > >
> > > > > > | Method | Avg. Pass Rate (Complexity) | Perfect Solve Rate (8/8 Correct) | Interpretation |
> > > > > > | :--- | :--- | :--- | :--- |
> > > > > > | **LPT (Baseline)** | **66.1%** | **36.7%** | Baseline (Easy)  |
> > > > > > | **Ours (Stage 1)** | **56.8%** | **25.5%** | Moderately Harder |
> > > > > > | **Ours (Stage 2)** | **38.4%** | **3.3%** | **Notably Complex** |
> > > > > >
> > > > > >
> > > > > > **Avg. Pass Rate:** The average percentage of correct answers over 8 attempts. Lower indicates higher difficulty.
> > > > > >
> > > > > > **Perfect Solve Rate:** The percentage of problems the model solved 8/8 times. **The drop to 3.3% highlights that Stage 2 data effectively produces notably more complex  problems**.
> > > > > >
> > > > > > (Figure 4a. The baseline is largely easier (green), while Stage 2 is predominantly complex (red/orange)).
> > > > > >
> > > > > > ---
> > > > > >
> > > > > >
> > > > > >
> > > > > > ### 2. Richness of CoT Trace Comparison
> > > > > >
> > > > > > We measure richness of the CoT trace by adding the average number of cognitive behaviors (Subgoals, Backtracking, Verification) present in each reasoning trace. (Figure 4b, Table 10)
> > > > > >
> > > > > > | Method | Richness CoT Trace (Sum of Avg ) | Relative Improvement |
> > > > > > | :--- | :--- | :--- |
> > > > > > | **LPT (Baseline)** | 0.65 | — |
> > > > > > | **Ours (Stage 1)** | 0.80 | +23% |
> > > > > > | **Ours (Stage 2)** | **1.99** | **+206% (~3x Richer)** |
> > > > > >
> > > > > > (see Table 10 for all details)
> > > > > >
> > > > > > Our Stage 2 data contains ***~3x more reasoning structures per trace***, **avg ~2 distinct cognitive behaviors per trace**, whereas the SoTA baseline is below 1.
> > > > > >
> > > > > > ---
> > > > > >
> > > > > > ### Re: Overall Novelty
> > > > > >
> > > > > > We emphasize that our work contributes the following advancements:
> > > > > >
> > > > > > * **Scale:** We overcome the bottleneck of previous SOTA methods (e.g LPT), allowing for the generation of **1M+ high-quality datapoints** comprising the entire post-training pipeline vs  30K. This is also not merely "more data" but a quantitative shift enabled by grounding MCQ generation on object-level metadata
> > > > > > * **Increased  Complexity:** As shown above, we significantly increase the difficulty of visual problem generation, reducing the rate of trivially solvable questions by **~10x (from 36.7% to 3.3%)** compared to the baseline.
> > > > > > * **Increased Richness of the CoT Trace:** We increase the richness of reasoning traces by **+206% (3x richer in cognitive behaviors)** compared to the baseline.
> > > > > > * **VLM Postraining Analysis:** Our comprehensive post-training analysis (Table 3, Section 4.2) reveals important scientific insights. Specifically, we show (i) online RL requires prior "skill teaching" via SFT, (ii) staged offline RL matches online RL performance with less compute, and (iii) high-quality SFT enables surprising cross-modal transfer (vision→audio +1.32%, vision→text +2.98%). **These findings have broader implications for VLM training**.
> > > > > > * **Resource contribution**: We are releasing one of the largest open vision-centric reasoning dataset to date (1M+ examples) supporting the full post-training spectrum. This is a significant resource for the community.
> > > > > > * **SOTA Performance:** The model fine-tuned on our data achieves SoTA performance among all open-data models.
> > > > > >
> > > > > >
> > > > > > ---
> > > > > >
> > > > > > **Note**:  Based on the above, we would appreciate it if the reviewer could clarify their definition of "novelty" in this context. We respectfully remind the reviewer of the ICLR Reviewer Guide [https://iclr.cc/Conferences/2026/ReviewerGuide]. *"Submissions bring value to the ICLR community when they convincingly demonstrate new, relevant, impactful knowledge (incl., **empirical**, theoretical, **for practitioners**, etc)."* **We believe our scaling framework and empirical results convincingly meet this standard**.

---

### Official Review · Reviewer_2af9 · 2025-10-31

**Soundness:** 2
**Presentation:** 3
**Contribution:** 2
**Rating:** 4
**Confidence:** 4

**Summary:**

The paper proposes Long Grounded Thoughts (LGT), a large-scale framework for compositional visual reasoning data synthesis. It integrates VLMs and LLMs in two stages: large-scale grounded question generation and composition hardening to build complex reasoning chains. Reasoning traces distilled from models are used to train Qwen2.5-VL-7B, which achieves strong gains on V*Bench, CV Bench, and MMStar-V, surpassing both open- and closed-source baselines.

**Strengths:**

1. The paper presents a clear and scalable framework for constructing large-scale compositional visual reasoning data. The two-stage process of grounded question generation and composition hardening is well structured and effectively implemented.

2. The experiments are comprehensive, covering multiple benchmarks and training paradigms including SFT, DPO, and GRPO, and they demonstrate consistent and meaningful improvements across tasks.

3. The paper contributes a large-scale reasoning dataset that enhances multimodal understanding and shows strong transferability to text and audio reasoning tasks.

**Weaknesses:**

1. The paper lacks clear methodological innovation and appears primarily engineering-oriented. While the proposed framework is well executed, its core idea builds on existing reasoning-chain distillation and data synthesis methods without introducing substantially new concepts.

2. The paper repeatedly emphasizes the limitations of current visual reasoning systems in the title and introduction, yet the experimental evaluation does not align with this focus. It omits reasoning-centric benchmarks such as VisuLogic and MathVision, which are crucial for demonstrating genuine reasoning improvements.

3. The paper compares its results mainly against general-purpose MLLMs (MiMo-VL) rather than task-specific reasoning baselines. This limits the fairness and informativeness of the comparisons, as it is unclear whether the observed gains stem from the proposed method or differences in data and model scale.

**Questions:**

Please refer to the Weaknesses section.

---

> ### Author Response · Authors · 2025-11-21
> **Response to Reviewer 2af9 - Part 1**
>
> We sincerely thank the reviewer for the helpful and constructive feedback, and for finding our "framework scalable " , our experiments “comprehensive” , and acknowledging that our work  “contributes a large-scale reasoning dataset that enhances multimodal understanding and shows strong transferability to text and audio reasoning tasks“
>
> Below, we address each concern in turn.
>
> ---
>
> ###  Concern 1: Methodological Innovation
>
> We thank the reviewer for this important question, which allows us to clarify our methodological contributions. While we agree that our framework prioritizes simplicity for scalability, we introduce **two concrete innovations** that solve fundamental bottlenecks in vision-centric reasoning data synthesis:
>
> **1\. Grounding-based synthesis breaks saturation.** Figure 2 demonstrates that LPT's caption-only approach saturates at \~100K datapoints, whereas our grounding-based method scales successfully to 1M+. This isn't just "more data". It's solving a **fundamental bottleneck** that prevented prior work from scaling.  The key insight is that conditioning generation on object-level metadata (bounding boxes \+ tags) prevents the collapse to repetitive questions that occurs when using only global captions. To the best of our knowledge, we are the first to demonstrate that grounding visual synthesis on spatial metadata enables breaking through this synthesis bottleneck.   ***Following the reviewers recommendation, we have also further added to the appendix a new analysis on diversity of the MCQ generated using our method vs LPT***. Please see section A.2 in the updated manuscript for additional visualization.
>
> **2\. Compositional hardening creates qualitatively different reasoning patterns.** Figure 4b and Table 10 show our data exhibits substantially higher frequencies of subgoal setting (0.55 vs LPT's 0.036), backtracking (0.68 vs 0.35), and verification (0.76 vs 0.26). These cognitive behaviors emerge from distilling from questions generated using our compositional hardening algorithm. Notably, they do not emerge at the same scale using the same distillation pipeline from LPT. Furthermore, figure 5 shows those skills are  **essential for effectively scaling online RL beyond 10K datapoints.**
>
> **Beyond methodology: Scientific insights for VLM post-training.**
>
> Our comprehensive analysis (Table 3, Section 4.2) reveals actionable findings that support the methodological innovations. We show specifically,  (i) online RL requires prior "skill teaching" via SFT on structured reasoning data (ii) staged offline RL (SFT→DPO) matches online RL performance with less compute, and (iii) high-quality SFT enables surprising cross-modal transfer (vision→audio \+1.32%, vision→text \+2.98%). These insights have direct implications for training future VLMs.
>
> **Resource contribution:** We are releasing one of the largest open vision-centric reasoning dataset to date (1M+ examples) supporting the full post-training spectrum. This is a significant resource for the community.
>
> We believe these contributions breaking the SFT scalability bottleneck through grounding, systematically generating MCQ that leads to non-linear reasoning via composition, and providing scientific insights into VLM post-training represent clear methodological advances beyond prior work.
>
> ---

---

> > ### Author Response · Authors · 2025-11-21
> > **Response to Reviewer 2af9 - Part 2**
> >
> > ### Concern 2: Missing Reasoning-Centric Benchmarks
> >
> > We are extremely thankful for  this excellent suggestion as it helped us clarify an important distinction in terminology and validate our data's generalization capabilities.
> >
> > First,  we understand the reviewer's concern about terminology. In our work,  by "vision-centric reasoning," we (and the broader vision community \[1,2,3,4,5\]) refer to tasks testing:  1\. visual search, 2\. 2D/3D spatial reasoning, 3\. fine-grained attribution, and 4\. scene understanding. **We do not refer to mathematical or logical reasoning**. Our work **intentionally focuses** on these vision-centric skills rather than visual math, which is why we evaluate on V\*Bench, CV-Bench, MMStar-V, etc. We emphasize this is desired and based on previous works[1,2,3,4,5\] that we follow for fair comparison.
> >
> > That said, **following your recommendation**, we evaluated our best  model finetuned on our data on **MathVision and VisuLogic**:
> >
> > | Model               | MathVision     | VisuLogic     |
> > |---------------------|----------------|---------------|
> > | Qwen2.5-VL-7B-Instruct | 49.57         | 11.5          |
> > | Ours (SFT and RL)      | 51.77  (2.2)    | 19.3   ( \+7.8)     |
> >
> > **Remarkably**, despite our data containing **no math questions**, we see meaningful improvements on both benchmarks.  This validates our data's generalization capability and demonstrates that strengthening core reasoning abilities transfers to out-of-domain tasks.
> >
> > We have  clarified this terminology distinction in the manuscript and will add the additional experiment on MathVision and VisuLogic to our supplementary material. If there are other specific benchmarks you would like us to evaluate, or any other distinction in terms of terminology, we would appreciate  the feedback,   please let us know.
> >
> > References:
> >
> > \[1\] Longperceptualthoughts: Distilling system-2 reasoning for system-1 perception.
> > \[2\] Are we on the right way for evaluating large vision-language models?
> > \[3\] Sft or rl? an early investigation into training r1-like reasoning large vision-language models.
> > \[4\] Cambrian-1: A Fully Open, Vision-Centric Exploration of Multimodal LLMs
> > \[5\] V\*: Guided Visual Search as a Core Mechanism in Multimodal LLMs
> >
> > ----
> >
> > ### Concern 3: Comparison Against General-Purpose vs. Task-Specific Baselines*
> >
> > We thank the reviewer for raising this important concern about baseline selection. We'd like to clarify our comparison strategy, experimental setting  and address potential misunderstandings.
> >
> > First, we emphasize **we do compare against multiple task specific  reasoning**  models in Table 2:
> >
> > 1. **LongPerceptualThoughts (LPT)**: Specifically designed for vision-centric reasoning with structured CoT traces (30K examples). We additionally extend their framework to generate 750K and 1M (see figure) to ensure the comparison is at the same scale, model size.
> > 2. **VLAA-Thinker**: Trained on 152K synthetic reasoning traces with both SFT and RL for multimodal reasoning
> > 3. **ReVisual-R1**: A reasoning-focused model trained on visual reasoning data
> >
> > These are all **task-specific reasoning models** trained on vision-centric synthetic reasoning data with similar objectives to ours.
> >
> > Furthermore, the reviewer notes it's "unclear whether the observed gains stem from the proposed method or differences in data and model scale." We believe  this is important to disentangle as we deliberate and carefully consider this:
> >
> > * **Controlled comparison with LPT**: We directly compare against LPT using the same base model (Qwen2.5-VL-7B) and similar data scale, showing our method achieves \+3.4 points improvement (0.716 vs 0.682). This isolates the effect of our synthesis method.
> > * **Scaling analysis**: Figure 2 explicitly compares scaling behavior between LPT and our method at multiple data scales (50K, 100K, 200K, etc.), showing our method continues to improve while LPT saturates. Importantly, this demonstrates our data quality advantage at the same scale both in terms of model and data.

---

> > > ### Author Response · Authors · 2025-11-21
> > > **Response to Reviewer 2af9 - Part 3 and Conclusion**
> > >
> > > **Additional context on open vs. closed data models:**
> > >
> > > **Why MiMo-VL is an important comparison point:** The reviewer is correct that MiMo-VL is a general-purpose MLLM. However, we include it as a key baseline for several reasons: (i) **SOTA performance on key benchmarks at the time of submission**: MiMo-VL-7B-RL achieved the highest published results on several of our target benchmarks (V\*Bench: 81.70, CV-Bench: 82.30) (2) **similar model parameters scale**: both use  7B as the base model (iii) **transparency limitation**: While MiMo-VL's training data is closed, it represents the performance ceiling that open-data approaches should aspire to match
> > >
> > > We acknowledge the comparison with closed-data models (MiMo-VL, GPT-4o, Claude 3.7) is for **reference only**.   It is not our intention to say **a  fine-tuned model resulting from a research artifact is better than a commercial production model at every task**.  Our primary claim is achieving the best results among vision-centric tasks finetuned on **open-data** strategies.   The fact that our model surpasses some closed-data models on 3/5 benchmarks demonstrates the strength of our approach but again is for reference only.    We have clarified this in the  caption of Table 2.
> > >
> > > ---
> > >
> > > ### Conclusion
> > >
> > > We emphasize that following the reviewer recommendation we have: 1. added new experimental results on the requested benchmarks. 2\. more explicitly categorized  baselines (task-specific reasoning vs. general-purpose), 3\. emphasize on the dedicated ablation comparing LPT vs. our method at matched data scales (Fig 1 and Tab 5), and 4\. clarified  which comparisons are "reference" vs. "primary" claims.
> > >
> > > We hope this clarification addresses the reviewer's concern. If there are specific task-specific baselines models the reviewer believes we should include, we would be grateful for those suggestions.

---

> ### Comment · Reviewer_2af9 · 2025-11-22
>
> Thank you for the rebuttal. I am going through the new results carefully. However, I still have some questions regarding the reported performance of Qwen2.5-VL-7B-Instruct. **According to the official technical report[1], Qwen2.5-VL-7B-Instruct achieves 25.1 on MathVision (25.4 according to OpenCompass Multi-modal Reasoning Leaderboard[2]), and the VisuLogic [3]paper reports a score of 26. The new results you provided in the rebuttal seem to differ from these official numbers. (49.57% vs. 25.1%, 11.5% vs. 26%)**
>
> It should be more fair to follow the evaluation protocol used in the official papers for these benchmarks.
>
> [1] Qwen2.5-VL Technical Report
>
> [2] VLMEvalKit: An Open-Source Toolkit for Evaluating Large Multi-Modality Models
>
> [2] VisuLogic: A Benchmark for Evaluating Visual Reasoning in Multi-modal Large Language Models

---

> > ### Author Response · Authors · 2025-11-22
> >
> > **We conducted all evaluations using the latest VLMEvalKit. Our results hold across all benchmarks**.
> >
> > We emphasize here that **we used up-to-date VLMEvalKit** for both MathVision and VisuLogic (as well as for all other benchmarks in the main section).
> >
> > Below, we further analyze where the discrepancy with the scores referred by the reviewer come from. Below are the details corresponding to each dataset.
> >
> > **TL;DR. Our results hold in all cases.**
> >
> >
> >
> >
> > **MathVision**
> > - We found that there exists a critical bug on multi-threading with MathVision that errors out when running evaluation. This is well-known, and has been previously reported in official VLMEvalKit repository (https://github.com/open-compass/VLMEvalKit/issues/898).
> > - Among multiple possible solutions to fix this bug, we chose to replace latex2sympy part of MathVision eval code with Huggingface `math_verify`. This not only fixes the bug but also leads to much better coverage and fairer evaluation, as latex2sympy (used in the problematic eval code) is known to often generate false negatives during answer parsing. As a result of reducing false negatives in the original metric, we see overall better scores across models.
> > - To more directly handle reviewer's concern, we additionally tested the original buggy setup by directly fixing the bug on latex2sympy. In this case, we get the following scores:
> >     - Qwen2.5-VL-7B: 25.4 (almost exactly reproduces the “reported” number)
> >     - Ours: **26.05** (**still outperforms**)
> > - Despite not being trained with latex-formatted answers (in fact, trained with a completely different format of `<answer>ANSWER</answer>`), our model still generalizes well to this specific format and keeps better performance than Qwen2.5-VL.
> > - In principle, even if an evaluation kit is widely used, if it has an apparent suboptimality (such as excessive false negatives), **we should use the improved version rather than naively accept the problematic results.**
> >
> >
> > **VisuLogic**
> > - Conversely for VisuLogic, we found that the paper repo’s eval (that the reviewer points to), is essentially using a different metric from VLMEvalKit. Specifically, the paper repo uses LLM judge with an API key (see function `extract_gpt` in VisuLogic-Eval github, `evaluation/eval_model.py`) to extract the answer from model-generated (possibly unstructured) CoT, while official VLMEvalKit employs rule-based matching on the boxed content (see function `VisuLogic_acc` in VLMEvalKit, `vlmeval/dataset/utils/visulogic.py`).
> > - Model generations often fail to follow the exact specified format. As the reviewer suggested, we aimed to stay consistent with the widely-used evaluation frameworks (unless there exists a direct blocker in the framework, such as bug).
> >
> > ---
> > ### Note :
> > While we show consistent improvements with LGT even on completely out-of-domain tasks such as on math & puzzle reasoning, **mathematical reasoning is not the goal, but a useful addition to our paper**. We focus on vision-centric tasks beyond math, as clearly stated in the abstract and intro (line 016, line 049, line 089).

---

> > > ### Author Response · Authors · 2025-11-27
> > > **Follow up Reviewer 2af9**
> > >
> > > Dear Reviewer 2af9,
> > >
> > > Given the approaching deadline, we remain committed to addressing any remaining questions you may have or would greatly appreciate you considering our response and reassessing the score.
> > >
> > > Thanks again for your follow up questions, your time and feedback!

---

### Official Review · Reviewer_yk5X · 2025-10-31

**Soundness:** 3
**Presentation:** 3
**Contribution:** 2
**Rating:** 4
**Confidence:** 4

**Summary:**

This paper introduces a scalable framework for constructing large-scale, vision-centric reasoning data, named Long Grounded Thoughts (LGT), which effectively integrates both large-scale generation and compositional complexity.

By fine-tuning Qwen2.5-VL-7B on the LGT dataset, the authors achieve state-of-the-art performance across five challenging vision-centric benchmarks.

In addition, the paper conducts comprehensive ablation studies comparing different post-training strategies, including supervised fine-tuning (SFT), direct preference optimization (DPO), and online reinforcement learning (GRPO), providing valuable insights into the effectiveness and efficiency of each method.

**Strengths:**

1. **Large-Scale and Well-Structured Data.**
The dataset is truly large-scale, containing over 1M multiple-choice questions (MCQs) paired with corresponding chain-of-thought (CoT) rationales, making it highly suitable for large-scale supervised fine-tuning. In addition, the authors also provide DPO preference pairs to support preference-based or reinforcement learning.


2. **Insightful Ablation Studies.**
The paper presents extensive and systematic comparisons across different post-training strategies (SFT, DPO, and GRPO), offering clear and valuable insights into the strengths and limitations of each approach.


3. **Strong Empirical Results and Generalization.**
Fine-tuning Qwen2.5-VL-7B on the proposed dataset achieves state-of-the-art performance on multiple vision-centric reasoning benchmarks and even surpasses GPT-4o and MiMo-VL-7B-RL in several settings. Moreover, the results demonstrate positive cross-modality transfer (from vision to text and audio), indicating improved general reasoning capability.

**Weaknesses:**

1. **Writing and Submission Quality**
   - **(1.1) Formatting and Completeness:**
     Although the conference does not enforce strict formatting rules, the paper’s presentation appears **rough and incomplete** for a formal submission. It uses **numerous single-column figures and tables** within the ICLR template, where visuals and text are often interleaved in a cluttered manner. Moreover, the main paper does **not even fill the full 9-page limit**, giving the impression of an unfinished submission.
   - **(1.2) Insufficient Data Illustration:**
     Despite claiming a **1M+ scale dataset**, the supplementary material provides **only two sample data examples**. This is far from sufficient to demonstrate the richness, quality, or diversity of the dataset, which undermines the paper’s credibility as a large-scale data contribution.
     Overall, these issues together make the paper’s **completion level appear quite low**.

2. **Content and Methodological Limitations**
   - **(2.1) Task Format Narrowness:**
     While the dataset is large, it is restricted to **multiple-choice questions (MCQs)** — a relatively simple task format. This **limits generalization** to open-ended or instruction-following reasoning, which are more representative of real-world multimodal tasks.
   - **(2.2) Limited Novelty in Core Techniques:**
     The data synthesis pipeline **largely builds upon LongPerceptualThought (LPT)** and similar reasoning-trace distillation approaches. The main contribution lies in **scaling and incremental improvements**, rather than proposing a fundamentally novel methodology.
   - **(2.3) Potential Bias and Lack of Human Audit:**
     As a dataset construction paper, it **lacks adequate human validation or auditing**. The authors explicitly acknowledge the presence of potential **societal biases** in the Ethics section but do **not propose any mitigation strategy**. This omission raises concerns about the reliability and fairness of the released dataset.

**Questions:**

1. **Generalization Beyond MCQs:**
   Since the proposed dataset is entirely **multiple-choice (MCQ)-based**, could models trained on LGT **generalize to open-ended reasoning or instruction-following tasks**? It would be helpful to know whether such transfer has been tested or if there are plans to include more open-ended data formats in the future.

2. **Impact on Original Model Capabilities:**
   After fine-tuning on the LGT dataset, did the model’s **original general capabilities** (e.g., text reasoning, captioning, or general multimodal understanding) **degrade in any way**? Please clarify whether you observed any trade-offs between improving vision-centric reasoning and maintaining the model’s broader skills.

---

> ### Author Response · Authors · 2025-11-21
> **Response to Reviewer yk5X - Part 1**
>
> We thank the reviewer for the helpful and constructive feedback, and for finding our dataset "truly large-scale," our ablation study insightful “offering clear and valuable insights into the strengths and limitations of each approach," and highlighting the strong generalization of our empirical results.
>
> We would also like to highlight that the reviewer's concerns have helped us strengthen our paper significantly.
>
>
>
> Below, we address each concern in turn.
>
>
> ---
>
> ### Formatting, Figures and Tables, and Data Illustration
>
> We appreciate the feedback and **sincerely apologize for the formatting issues around figures and tables**, and for only providing 2 qualitative examples in the supplementary material. This was an oversight on our side. **Following your recommendation**, we have now  **restructured several tables and figures** and resolved the formatting issues. We have also **incorporated additional qualitative results** into the main paper. Please **see Figures 3 and 5 in the main text**, and **Figures 7 and 8 in the supplementary material**.
>
>
> Additionally, we have provided a **significantly larger set samples for your evaluation** ( we have uploaded a new suppl material file) . Please kindly let us know if you would like to see more examples, or if there is any formatting issue that is still a concern.
>
> ---
>
> ### Task Format Narrowness:
>
> We appreciate this important question. While we acknowledge the MCQ limitation (noted in our limitations section), **we have specifically tested generalization to open-ended tasks to address this concern**. Below we show results on  the Needle in Embodied Haystack (NiEH) benchmark—an ***open-ended***, ***multi-frame*** VQA task completely different from our training data:
>
> | Model                      | Score  |
> |----------------------------|--------|
> | Qwen2.5-VL-7B-Instruct     | 47.55  |
> | + VLAA-thinking                      | 47.85  |
> | + Ours (SFT best)          | 48.24  |
> | **+ Ours (SFT + RL \*best)**     | **56.34** |
>
> This demonstrates that the reasoning skills learned from MCQs transfer effectively to complex, open-ended scenarios, directly addressing the reviewer's concern about task format narrowness. We emphasize again 1) this is an open-ended, free-form VQA generation task (not MCQ), 2) requires multi-frame reasoning, and 3)  our training used only single-frame MCQs. Result and does not contained any embodied VQA type question.
>
> Following the reviewer recommendation, we ***additionally run experiments on MathVision and VisuLogic***.  Results are shown below.
>
>
> | Model               | MathVision     | VisuLogic     |
> |---------------------|----------------|---------------|
> | Qwen2.5-VL-7B-Instruct | 49.57         | 11.5          |
> | Ours (SFT and RL)      | 51.77  (2.2)    | 19.3   ( +7.8)     |
>
>
> **Remarkably**, despite our data containing ***no math questions***, we see meaningful improvements on both benchmarks.
>
> This additionally  validates our data's generalization capability and demonstrates that strengthening core reasoning abilities transfers to out-of-domain tasks open ended data.
>
> We also emphasize that despite MCQs appearing relatively simple, ***no existing dataset (either MCQ or open question)  provides vision-centric reasoning with complex structured reasoning traces at this scale (see updated Table 1)***. Our work fills this critical gap. Moreover, as you noted, our results also demonstrate "positive cross-modality transfer (from vision to text and audio), indicating improved general reasoning capability."

---

> > ### Author Response · Authors · 2025-11-21
> > **Response to Reviewer yk5X - Part 2**
> >
> > ### Limited Novelty in Core Techniques:
> >
> > We appreciate the reviewer highlighting the importance of scalability and for recognizing that our ablation studies offer 'clear and valuable insights into the strengths and limitations of each approach’. Here, we'd like to clarify that while we emphasize simplicity as a design principle for successful scaling, our work is not merely incremental. Our work introduces two key methodological contributions that as the  reviewer noted that are important for the community:
> > 1. ***Grounding-based synthesis breaks saturation***. Figure 2 demonstrates that LPT's caption-only approach saturates at ~100K datapoints, whereas our grounding-based method scales successfully to 1M+. This isn't just "more data". It's solving a fundamental bottleneck that prevented prior work from scaling. **Following your recommendation, we have also further added to the appendix a new analysis on diversity of the MCQ generated using our method vs LPT**. Please see section A.2 in the updated manuscript for additional visualization and details.
> > 2. ***Compositional hardening creates quantitatively different reasoning patterns***. Figure 4b and Table 10 show our data exhibits substantially higher frequencies of subgoal setting (0.55 vs LPT's 0.036), backtracking (0.68 vs 0.35), and verification (0.76 vs 0.26). These cognitive behaviors emerge from distilling data on MCQ resulting from our composition algorithm and are useful for effective online RL (Figure 5 shows GRPO fails to elicit  them unless injected during SFT).
> >
> > ***Beyond methodology***, our comprehensive post-training analysis (Table 3, Section 4.2) reveals important scientific insights that align with your  observation about our work providing  'clear and valuable insights’. Specifically, we show   (i) online RL requires prior "skill teaching" via SFT, (ii) staged offline RL matches online RL performance with less compute, and (iii) high-quality SFT enables surprising cross-modal transfer (vision→audio +1.32%, vision→text +2.98%). ***These findings have broader implications for VLM training***.
> >
> > Finally, ***we are releasing one of  the largest open vision-centric reasoning dataset (1M+ examples) supporting the full post-training spectrum***. We believe this is  a significant resource contribution to the community.
> >
> > ---
> >
> >
> > ### Human Validation and Bias Mitigation:
> >
> > We appreciate this important concern. Our pipeline currently incorporates: (1) automated LLM-based verification for logical consistency (2) semantic deduplication (3) grounding in human-annotated DOCCI captions. We welcome the reviewer's guidance on specific methodologies or bias categories to evaluate on a specific subset.
> > Would the reviewer recommend specific evaluation protocols or bias taxonomies from the literature that would strengthen our validation? If the reviewer has a concrete  suggestion in mind, in a particular stage of the synthesis, please let us know and, we will make the best effort to provide a preliminary analysis  during rebuttal .

---

> > > ### Author Response · Authors · 2025-11-21
> > > **Response to Reviewer yk5X  - Response to Specific Questions**
> > >
> > > ## Response to Specific Questions:
> > >
> > > ---
> > >
> > > ### Generalization Beyond MCQs
> > >
> > > This is a very good question. We evaluate our best models on **Needle in the Embodied Haystack (NiEH)**, an **open-ended embodied VQA benchmark** (not MCQ). Please see results on the section on Task Narrowness.  Critically  1) this is open-ended free-form generation, 2) requires multi-frame reasoning, yet (3) we trained only on single-frame MCQs question. Our purely MCQ, single-frame training data generalizes to open-ended, multi-frame embodied VQA with **+8.8 points** improvement over the base model. Similarly, we observe **notable gains on MathVision and VIsuLogic** in spite of our data ***not containing a single math problem or diagram***. Notice that MathVision is a mixture between MCQ and ***open ended questions***.
> > >
> > > We emphasize that we design the dataset as MCQ on purpose due to the benefits it brings for SFT verification, offline RL synthesis, and online RL reward computation.
> > >
> > > ---
> > >
> > >
> > > ### Impact on Original Model Capabilities
> > >
> > > This is a very insightful question. We tested this thoroughly because we had the same concern while developing the dataset.   As shown below, fine-tuning on our data **improves** (rather than degrades) ***out-of-domain capabilities***:
> > >
> > >
> > >
> > > **MMLU-Pro (Text-Only Reasoning)**
> > > | Model                   | Acc   |
> > > |-------------------------|-------|
> > > | Qwen2.5-VL-7B-Instruct  | 47.15 |
> > > | **+ Ours (SFT)**        | **50.13** |
> > >
> > >
> > >
> > > **Omni-Modal Results (Qwen2.5-Omni-7B)**
> > > | Model              | MMAU-Sound | MMAU-Music | MMAU-Speech | MMLU-Pro |
> > > |--------------------|------------|------------|-------------|----------|
> > > | Baseline           | 76.77      | 67.33      | 68.90       | 47.00    |
> > > | + Virgo            | 64.20      | 59.30      | 64.30       | 39.21    |
> > > | + LPT              | 76.63      | 67.56      | 66.93       | 48.74    |
> > > | **+ Ours (SFT+DPO)** | **77.75** | **70.35** | **69.23**   | **51.07** |
> > >
> > > In all cases, our data outperforms both the base model and comparable open-source datasets, demonstrating that our vision-centric reasoning data **enhances rather than degrades** general capabilities across modalities. We believe that this is a very important and interesting finding.
> > >
> > > ---
> > >
> > > ### Conclusion
> > >
> > > We hope these clarifications address your concerns and appreciate your insightful comments. We respectfully ask the reviewer to reconsider the rating in light of these changes.  Please  let us know if this satisfies your concern or whether there is anything else you would like us to explain or any other experiment you would like us to conduct.

---

> ### Author Response · Authors · 2025-11-27
> **Follow up reviewer yk5X**
>
> Dear Reviewer yk5X,
>
> We submitted a comprehensive response one week ago with extensive new experiments and a revision of the manuscript PDF (changes in blue)  directly addressing your concerns, including restructured figures and tables, additional qualitative examples (new suppl. material zip file) , OOD generalization results on NiEH, MathVision, and VisuLogic, diversity analysis comparing our method to LPT, and validation that our data enhances rather than degrades general capabilities across modalities.
>
> Given the approaching deadline, we remain committed to addressing any remaining questions you may have or would greatly appreciate you considering these contributions and reassessing the score.
>
> Thanks again for your time and feedback

---

### Official Review · Reviewer_U8gg · 2025-11-07

**Soundness:** 2
**Presentation:** 3
**Contribution:** 3
**Rating:** 6
**Confidence:** 4

**Summary:**

This paper is notable as a resource contribution but less convincing as a methodological advance. On the positive side, the authors curate a large-scale (≈1M) vision–language reasoning corpus constructed via a two-stage pipeline: (i) grounded single-hop MCQs generated from image captions and object-level metadata, and (ii) “composition hardening” that merges simpler items into multi-hop questions. They further attach chain-of-thought rationales distilled from stronger models. The dataset’s scale, grounding, and intended openness are likely to be valuable to the community and could enable more rigorous benchmarking of compositional visual reasoning.
However, the algorithmic core is insufficiently specified and theoretically underdeveloped. The composition procedure is implemented through prompt heuristics without a formal definition of difficulty, verifiability guarantees, or complexity controls. The reasoning-chain distillation relies on very large teacher models and ad-hoc filtering, raising concerns about reproducibility, error propagation in rationales, and the extent to which the chains reflect genuine reasoning rather than stylized verbosity. Empirically, several critical ablations are missing (e.g., training without CoT, without composition, or with open-ended targets), and the exclusive MCQ training leaves generalization to free-form VQA underexplored. Finally, some headline comparisons to proprietary systems are presented without clear apples-to-apples protocols, which weakens the strength of the claims.

**Strengths:**

- Scale and accessibility. Curates ~1M image–question pairs with aligned reasoning traces—an order-of-magnitude jump over prior open corpora (e.g., LPT ~30k). The inclusion of stepwise solutions and preference labels makes the resource directly usable for SFT, reward modeling, and RL without bespoke tooling.

- Grounded synthesis that resists mode collapse. Stage-1 leverages object-level signals (boxes/tags) to anchor prompts to specific regions, empirically sustaining question diversity as the corpus scales. Stage-2 composes single-hop items into multi-hop queries, injecting controlled compositionality rather than merely elongating prompts.

- Data-driven performance lift at small scale. A 7B VLM fine-tuned on this corpus surpasses prior open baselines and matches/edges select closed 7B systems on multiple vision-reasoning suites, indicating that targeted reasoning supervision can trade off against parameter count. The SFT→DPO regimen reproduces most of online-RL gains at substantially lower complexity.

- Structured rationale profiling. Quantifies incidence of subgoaling, backtracking, and self-verification in the chains, with higher frequencies than earlier datasets. While not causal, the analysis supports that traces encode non-trivial procedural structure likely to benefit compositional reasoning training.

**Weaknesses:**

- The pipeline is a stack of heuristics with no formal objective or guarantees. “Composition hardening” is described informally; there is no principled difficulty measure, constraint set, or verification proof that the composed item requires multi-step reasoning rather than adding length.

- Stage-1 (caption-->MCQ) largely reprises LPT with object tags/boxes to steer diversity; Stage-2 composition is another prompt-level heuristic; rationale “extension” mirrors prior VLM-->LLM distillation. Crucially, there is no ablation isolating Stage-2’s marginal value over scaled single-hop data; if gains are driven by volume rather than composition, the claimed advance weakens.

- Chains are seeded by a VLM and elongated by an LLM, which risks early errors and producing stylized verbosity. Without human audits or automatic logic checks, it is unclear whether chains exhibit valid knowledge.

- Data synthesis depends on frontier teachers (e.g., 235B models) and an LLM verifier plus regex constraints. This raises reproducibility and availability concerns and suggests sensitivity to prompt/decoder settings.

- Comparisons emphasize selective wins over closed 7B systems but do not present apples-to-apples protocols (prompting, vision enablement, few-shot context). Missing baselines include: (i) no-CoT training, (ii) Stage-1-only vs. Stage-1+2, (iii) larger open models fine-tuned on a subset, and (iv) human upper bounds.

- Robustness and generalization remain unclear. Training data is synthetic, MCQ-only, and anchored to a single caption source, risking domain/style narrowness. There is limited evidence for transfer to free-form tasks, diagrams, or out-of-distribution imagery.

**Questions:**

- How exactly is composition hardening done, and how do you verify it truly requires multi-step reasoning?

- What is the performance gap with identical data without CoT, with short CoT, and with full CoT?

- What were the exact GPT-4/Claude settings (vision, prompt, shots, temperature), and do results hold under a standardized protocol?

- What are filter rejection rates and a human-audited error rate/taxonomy for QAs and CoTs?

- How much does performance drop when replacing 235B/671B teachers with accessible open models (13B–70B)?

---

> ### Author Response · Authors · 2025-11-21
> **Response to reviewer U8gg**
>
> We sincerely thank the reviewer for their thorough evaluation and insightful feedback. We particularly appreciate the recognition of our work's scalability, comprehensive experiments, and contribution to multimodal understanding.
> The reviewer's concerns have helped us strengthen our paper significantly.
>
> Below, we address each point with the requested additional experiments and clarifications.
>
> ---
>
>
> ### Concern 1: Formal Objectives and Guarantees
>
> We appreciate this important observation. **The reviewer is absolutely correct** that our work prioritizes empirical scalability over formal guarantees. We'd like to clarify our contribution and positioning:
>
> **Our contribution:** Rather than proposing a theoretically-grounded algorithm, we provide a **transparent, reproducible recipe** that addresses two concrete, previously unsolved bottlenecks in vision-reasoning data synthesis:
> 1. **Scalability bottleneck (Stage 1):** We **empirically demonstrate** that grounding-based synthesis breaks the saturation that limited prior work to ~100K examples (Figure 2). This is not merely "more data" but a quantitative shift enabled by grounding MCQ generation on object-level metadata.
> 2. **Complexity bottleneck (Stage 2):** We show that compositional hardening creates **measurably different reasoning patterns** (Figure 4 and a new Table 10 on the appendix shows subgoal setting: 0.12 → 0.55, backtracking: 0.35 → 0.68, verification: 0.33 → 0.76). In fact, most existing open source dataset (see updated Table 1 in the paper) lack those  behaviors which  our experimental analysis on VLM postraining shows to be essential to scale  online RL further.
>
> **On formal guarantees:** We agree that formal difficulty measures and verification proofs would strengthen the work. However, we believe establishing formal frameworks for vision-reasoning synthesis at this scale remains an open research question beyond the scope of this contribution. Our work takes a necessary first step by demonstrating **what works empirically** at scale, which we hope will inform future theoretical frameworks.  We have clarified in the paper (limitations) that our approach is empirically-driven and positioned as a scalable recipe rather than a formally-guaranteed algorithm.
>
> ---
>
> ### Concern 2: Stage 1 and Stage 2 Value
> Thank you for this critical question. We have conducted extensive ablations to isolate each stage's contribution:
>
> --
>
> ### Stage 1 Value: Breaking the Saturation Bottleneck
>
> **Quantitative evidence:**  Figure 2 demonstrates that LPT's caption-only approach saturates at ~100K datapoints (performance plateaus)  Our grounding-based method successfully scales to 1M+ with **continued performance gains** . We believe this is not simply  "reprising LPT" but solving a fundamental bottleneck
>
> **New diversity analysis**:  Following your recomendation, we have conducted a new diversity analysis on the MCQ synthesized by our approach vs LPT. ***We have added this to the supplementary material(Section A.2)*** where we have further show two visualization plots. Specifically, we sample $N=1000$ MCQ questions generated from three shared seed images for both LGT (Ours) and LPT (Baseline). We encode these questions using SentenceBERT and estimate their semantic spread and average pairwise cosine similarity in the embedded space.  Here, we summarize the results:
>
> - ***Spread:*** LGT yields a  ***3.2x wider*** distribution (mean Euclidean distance to centroid) than the baseline.
> - ***Information Density:*** LGT significantly reduces redundancy, with an average pairwise cosine similarity of 0.61 compared to  0.82 for LPT.
>
> These results suggest that object-level metadata prevents MCQ synthesis collapse and maintains diversity at scale. We refer the reviewer to Section A.2 for more details on the newly added experiment.
>
> --
> ### Stage 2 Value: Adding Compositional Complexity
>
> To  address this concern, first we compute the cognitive behaviours elicit by stage 1 vs stage 2. Those are quantified based on the methodology presented in Ghandi et al and Liao et al.  These metrics confirm Stage 2 successfully induces highly notable improvements.
>
> | Behavior | Stage 1 Only | Stage 1+2 | Improvement |
> |----------|--------------|-----------|-------------|
> | Subgoal setting | 0.12 | 0.55 | +358% |
> | Backtracking | 0.35 | 0.68 | +94% |
> | Verification | 0.33 | 0.76 | +130% |
>
> Furthermore, we isolate and quantify the gains of adding stage 2 in the following experiment.  Notably, stage 2 adds **+3.15% on V*Bench** (a visual search benchmark requiring search and  reasoning) and ***improves 3/4 evaluated*** benchmarks.
>
> | Training Strategy | V*Bench | CVBench | MMVP | RealWorldQA | MMStar-V |
> |-------------------|---------|---------|------|-------------|-----------|
> | SFT (Stage 1 Only) | 79.05 | 80.60 | 73.67 | 65.49 | 64.27 |
> | SFT (Stage 1) + RL | 80.10 | 81.51 | 74.00 | 66.14 | 64.40 |
> | **SFT (Stage 1+2) + RL** | **83.25** | **82.28** | 72.33 | **68.76** | **66.27** |

---

> > ### Author Response · Authors · 2025-11-21
> > **Response to reviewer U8gg (Part 2)**
> >
> > ### Concern 3: Dependence on Large Teachers
> >
> > Thank you for raising this important concern. We'd like to clarify our teacher model strategy:
> > - **Stage 1 (~750K examples):** Uses accessible models (Qwen2.5-VL-7B for VLM, DeepSeek-R1-32B for LLM)
> > - **Stage 2 (~250K examples):** Uses frontier models (Qwen2.5-VL-72B, R1-671B, Qwen3-235B)
> >
> > Notably **the bulk of our data (75%) uses models within reach of most researchers**.  Stage 2's frontier models produce the richest reasoning chains for the hardest problems.  We will release **both** datasets. Our Stage 1-only ablation (Table above) shows that removing Stage 2 entirely (thus removing all frontier model dependence) still achieves strong performance (80.10  on V*Bench vs. 83.25 with Stage 2). This suggests **Stage 1 +DPO alone provides substantial value** even without access to the largest models.  We emphasize we will be releasing both Stage 1 and Stage 2 data separately, enabling the community  to make informed choices.
> >
> > ---
> >
> > ### Concern 4: Closed-Source Comparisons and Baselines
> >
> > We appreciate the reviewer pointing out the need for clarity here. We have revised the paper to better frame these comparisons:  Specifically, we emphasize that  our model achieves **best results among open-data approaches** on vision-centric reasoning benchmarks. As a secondary observation, we emphasize that our model surpasses some closed-data models (MiMo-VL-7B-RL) on 3/5 benchmarks, demonstrating the strength of high-quality open data. We do not claim superiority over production systems; rather, we demonstrate that **open data can be competitive** in ***narrow vision-centric domains***. Specifically,in the revised manuscript on table 2 caption we  now state: "Closed-source results are for reference only and taken from technical reports". Also in section 4.1 we clarify  "Our primary claim is achieving best results among open-data strategies".
> >
> > We believe emphasizing the  framing respects the reviewer's concern while highlighting our contribution to open-source VLM development. Please let us know if you would like us to adjust something else here.
> >
> > ---
> >
> > ### Concern 5: Validation Without Human Audits
> >
> > This is an excellent point. While human audits would be ideal, we emphasize we implemented a **comprehensive automatic validation pipeline** throughout data generation. We specifically have several **multi-stage automatic checks:**
> > 1. **Stage 1 MCQ generation:**
> >    - LLM verifier  validates logical correctness and ground-truth answer validation against caption
> >    - Format consistency checks
> >    - Semantic similarity filtering (removes near-duplicates)
> > 2. **Simple CoT generation:**
> >    - Answer consistency check (CoT must reach correct answer)
> >    - Format validation (<think> tags)
> > 3. **Thought expansion:**
> >    - Re-validation of final answer after expansion
> >    - Regex-guided decoding to prevent bad words.
> >
> > **Indirect human validation:** Our strong empirical results across diverse benchmarks (including out-of-domain transfer) provide indirect evidence that the automatic validation is effective.  Following your recomendation, we have emphasized this part in the main manuscript as it was previously mentioned very briefly.
> >
> > While we agree that systematic human evaluation would further strengthen confidence, we defer explicit  human audits to  future work.
> >
> > ---
> >
> > ### Concern 6: Robustness and Generalization
> >
> > We greatly appreciate this concern, as it motivated us to conduct extensive out-of-distribution evaluations. The results **strongly support** the generalization of our data:
> >
> > --
> >
> > ### Evidence 1: Open-Ended VQA (Not MCQ, out of distribution multi-frame embodied VQA reasoning)
> > **Needle in Embodied Haystack (NiEH)** - multi-frame, open-ended VQA:
> >
> > | Model | Score |
> > |-------|-------|
> > | Qwen2.5-VL-7B-Instruct | 47.55 |
> > | + **Ours (SFT + DPO)** | **56.34** (+8.79) |
> >  Despite training exclusively on MCQs, our model shows substantial improvement on open-ended  embodied VQA reasoning.
> >
> > --
> >
> > ### Evidence 2: Math Reasoning (Unseen Domain all our images are natural images as the reviewer pointed)
> >
> > | Model | MathVision | VisuLogic |
> > |-------|------------|-----------|
> > | Qwen2.5-VL-7B-Instruct | 49.57 | 11.5 |
> > | + Ours (SFT + DPO) | **51.77** (+2.2) | **19.3** (+7.8) |
> >   Core reasoning abilities transfer to mathematical domains despite no math data synthesized.
> >
> > --
> >
> > ### Evidence 3: Cross-Modal Transfer
> >
> > **Text-only reasoning (MMLU-Pro):**
> > - Base: 47.15 → **Ours: 50.13** (+2.98%)
> >
> > **Audio reasoning (MMAU on Omni-7B):**
> > - Sound: 76.77 → **78.30** (+1.53)
> > - Music: 67.33 → **70.20** (+2.87)
> > - Speech: 68.90 → **69.23** (+0.33)
> >
> > Vision-centric reasoning improves text and audio reasoning, suggesting our data teaches transferable cognitive skills.

---

> ### Author Response · Authors · 2025-11-21
> **Response to reviewer U8gg (Part 3) - Responses to Specific Questions**
>
> ## Responses to Specific Questions
>
> ---
>
> ### Q1: How exactly is composition hardening done, and how do you verify it requires multi-step reasoning?
> We apologize for not defining what we specifically mean by muti-step reasoning as this term may be overloaded in the literature.  We define multi-step reasoning as a reasoning trace that decomposes the original problem into manageable sub-steps to reach the final answer.
>
> **Compositional Hardening Process:** Given K problems from Stage 1, an LLM is tasked with creating a single, creative, and challenging question that **ideally**  requires solving the multiple sub-questions to reach the final answer (see Appendix A.3 for prompt details).
>
> **Verification:** While we agree formal proof is ideal, we use established cognitive behavior metrics (Gandhi et al., 2025) to quantify subgoal settings (e.g. the  reasoning trace explicitly decomposes the problem into subgoals or steps)   :
>
>
> | Metric | Stage 1 | Stage 2 | Increase |
> |--------|---------|---------|----------|
> | Subgoal setting | 0.12 | 0.55 | +358% |
> We believe the substantial increase in subgoal setting in our data due to stage 2  inderectly indicates multi-step decomposition is required for those problems.
>
>
> We thank the reviewer for this question, and have clarified in the manuscript what we specifically mean by multi-step reasoning. Please if you have any other comments, or want us  to run a specific experiment let us know.
>
>
> ---
>
> ### Q2: Performance gap without CoT, with short CoT, and with full CoT?
>
> Following the reviewer’s recommendation, we conducted an experiment training for 1 epoch on 750K Stage 1 datapoints. We explicitly constructed three datasets with identical MCQs but controlled reasoning traces: **No CoT** contains only direct responses, **Short CoT** includes simplified traces without cognitive behaviors, and **Full CoT** contains our full reasoning annotations.   We also added a prompted baseline where we instruct Qwen2.5-VL in the system prompt to “think,”. We evaluate  all models on two challenging visual reasoning benchmarks focused on visual search and 2D/3D spatial reasoning.
>
> | Model | CV-Bench | V\*Bench |
> |-------|----------|----------|
> | Qwen2.5-VL-7B-Instruct | 0.740 | 0.485 |
> | + Thinking prompt (NO SFT) | 0.754 | 0.551 |
> | + SFT No CoT | 0.787 | 0.583 |
> | + SFT Short CoT | 0.714 | 0.540 |
> | + SFT Full CoT | 0.813 | 0.597 |
>
> Finetuning on **Full CoT** traces yields the strongest performance across both benchmarks. This result is obtained **without any RL training**, which (as shown in Figure 5) further benefits from injecting non-linear behaviors during finetuning.   Interestingly, finetuning on **Short CoT** performs worse than **No CoT**, letting open interesting avenues for further research.
>
> We thank the reviewer for this question, and would like to emphasize we will release all version of the stage 1 SFT dataset, as they may be useful to the community in different ways.
>
> ---
>
> ### Q3: GPT-4/Claude settings and standardized protocol?
>
> We appreciate the need for clarity. As noted in our response to Concern 4, we emphasized that  Closed-model results are **reference-only** obtained  from technical reports. Our **primary claim** is best performance among **open-data approaches**. We do not claim apple-to-apple comparisons with closed systems as by is nature is very difficult to do wrt to a close source system, and developers of closed models typically report the best they can achieve on those models.  We have clarified this explicitly in Table 2 caption and Section 4.1.
>
> ---
>
> ### Q4: Filter rejection rates and error taxonomy?
> We appreciate this question but are a bit uncertain what experiment the  reviewer specifically has in mind. Could the reviewer clarify what specific error taxonomy would be most valuable to them? and for which pipeline stage/subset of data would this analysis be most informative?
>
> ---
>
> ### Q5: Performance with smaller teachers (13B–70B)?
>
> Our **Stage 1-only ablation** (see Concern 2 response) provides a proxy to answer this question.  Stage 1 uses 8B VLM + 32B LLM and alone achieves 79.05 on V*Bench. Stage 1+2 achieves 83.25 on V*Bench. Therefore, removing Stage 2 (which uses 235B/671B teachers) costs ~4 points on V* bench, suggesting: i)   Stage 1 with accessible models provides substantial value (79.05 vs 75.39 base).
>
> ---
>
> ###  Larger models in a small subset
>
>
> The reviewer also suggested finetuning  larger open models fine-tuned on a subset . Below, we show the results of the **Qwen2.5-VL-32B** model finetuned (SFT only) on **500K** datapoints of our SFT data.  The results suggest our data transfer also to larger models.
>
>
> | Model        | Avg   | CV-Bench | V∗ Bench | MMVP  | MMStar-V |
> |--------------|-------|----------|----------|-------|----------|
> | Base Model (32B)     | 0.689 | 0.743    | 0.587    | 0.741 | 0.684    |
> | SFT Only +  500K (Ours)   | **0.735** | **0.854**    | **0.623**    | **0.760** | **0.704**    |

---

> > ### Author Response · Authors · 2025-11-21
> > **Response to reviewer U8gg -  Conclusion**
> >
> > ### Conclusion
> >
> > We thank the reviewer again for their thoughtful evaluation. The concerns raised have led us to: ***1)*** conduct extensive new ablations isolating Stage 1 and Stage 2 value. ***2)***  demonstrate strong out-of-distribution generalization (embodied QA, math, cross-modal). ***3)***  clarify our positioning relative to closed-source systems. ***4)*** highlight better our  comprehensive automatic validation pipeline. ***5)*** provide analysis on the teacher model and show generalization of our data to also bigger models.
> >
> > We believe these additions substantially strengthen the paper and address the reviewer's concerns. **We hope the reviewer will consider these contributions when reassessing the score**, as we believe our work makes significant empirical advances in open-source vision-reasoning data synthesis at scale. If any concerns remain, we are happy to provide additional experiments or clarifications.

---

> > > ### Author Response · Authors · 2025-11-27
> > > **Follow up with reviewer U8gg**
> > >
> > > Dear Reviewer U8gg,
> > >
> > > We submitted a comprehensive response one week ago with extensive new experiments directly addressing your concerns, including  new ablation studies,  OOD generalization results, additional analysis, and also new version of the manuscript with updates marked in blue.
> > >
> > > Given the approaching deadline, we remain committed to addressing any remaining questions you may have or would greatly appreciate you considering these contributions and reassessing the score.
> > >
> > > Thanks again for your time and feedback.

---

### Meta-Review · Area_Chair_S9vo · 2026-01-01

**Summary:**

Recent multimodal reasoning relies heavily on private data and lacks systematic methods for creating large-scale reasoning datasets centered on vision. Therefore, this paper proposes a two-stage synthetic framework covering diverse skills and difficulty levels. This framework generates scale and integrates complexity, constructing over one million verifiable visual problems with preference and instruction data. Fine-tuning Qwen2.5-7B-VL on this dataset enables the model to surpass open-domain state-of-the-art (SOTA) performance on various vision benchmarks and outperform some closed-domain models on certain tasks. Additionally, transfer learning can be applied to text and audio reasoning.

This paper proposes visual reasoning learning based on large-scale synthetic data. The additional experiments and explanations reinforced by Rebuttal are commendable. The experimental demonstration that data constructed in MCQ format can generalize to non-MCQ settings (e.g., free-form tasks) is particularly valuable regarding dataset design.

However, fundamental concerns remain unresolved. The approach essentially extends existing LPT or chain-of-reasoning distillation/data synthesis methods with limited methodological novelty or learning innovation. Furthermore, the approach still relies heavily on MCQs and a single caption source, leaving generalizability to broader real-world settings, such as charts and out-of-distribution images, weakly substantiated. Therefore, while the experimental validity is confirmed to a certain extent, the current level of completion is insufficient in terms of clarity of scientific contribution and generalizability. The AC recommends rejecting this paper.

**Reviewer Concerns:**

The reviewers raised the following concerns:

The work remains limited to extensions of LPT and existing reasoning-chain distillation/data synthesis. It lacks "methodological novelty or learning innovation." The principles of "enhanced synthesis/Stage 2" and the description of the inference process are unclear. There is insufficient differentiation between Stage 1 vs. Stage 1+2, the presence or absence of CoT, and quantity vs. composition. There is also insufficient verification of which elements were effective. Despite concerns about error propagation, redundancy, and response bias from VLM to LLM, manual auditing, automated logic checks, and bias mitigation are weak. There is heavy reliance on external dependencies (massive teacher model, verifier, and regular expression constraints), and reproducibility is unclear, including sensitivity to prompt and decoder settings. It lacks reasoning-centric benchmarks (e.g., VisuLogic/MathVision), and comparisons lack unified protocols. Crucial controls, such as task-specific baselines or human upper bounds, are absent. The MCQ-centric approach with a single caption source limits domain and style coverage. There is weak transfer evidence for free-form text, charts, or out-of-distribution images. The presentation is rough, the sample presentation is sparse, and the quality and diversity of the large-scale data are not persuasive.

The authors' rebuttal provides additional experiments and explanations that appear to address many of the above concerns.
However, concerns remain regarding the lack of methodological novelty or learning innovation. The focus on MCQs persists, as does the weak transfer evidence for free-form text, charts, and out-of-distribution images.

**Reviewer Scores:**

Regarding methodological novelty and innovation in learning, the authors state that grounding-based synthesis overcomes saturation and constructive reinforcement produces qualitatively different inference patterns. While these certainly provide counterarguments, it is difficult to argue that they "fully resolve concerns about methodological novelty." The primary claim presented is that "by skillfully reconfiguring existing elements (LPT systems, distillation, RL) to avoid bottlenecks, scaling yielded performance improvements." As new principles, guarantees, or generalized algorithmic insights, this remains relatively weak.

In response to the concern that the dataset is limited to multiple-choice questions, the authors applied the best model to Needle in the Embodied Haystack (NiEH), MathVision, and VIsuLogic, confirming performance improvements. These results demonstrate the versatility of the proposed method, and the AC results are also considered excellent. However, the analysis of why and to what extent the MCQ format constraint enables generalization is insufficient. The representativeness and limitations of free-response and instruction-following tasks are not fully explained.

Multiple reviewers raised the above concerns. For reviewers who initially gave negative scores, the justification for moving toward an accept decision seems somewhat weak.

---

### Decision · Program_Chairs · 2026-01-26

Reject